# Real environment obstacle circular edge expansion design robot path planning based on ant colony algorithm

Feng Li[1,2,3]*, Maoya Yang[2], Seong-Nam Jo[2], Young-Chul Kim[2], Ziang Lyu[2]

**1** Department of Intelligent Manufacturing, Zhengzhou University of Economics and Business, Zhengzhou, Henan, China, **2** Department of Mechanical Engineering, Kunsan National University, Gunsan, Jeollabuk, Republic of Korea, **3** Henan Province Engineering Research Center of Multimodal Perception and Intelligent Interaction Technology, Zhengzhou, Henan, China

* lifeng@kunsan.ac.kr

## Abstract

With the development of artificial intelligence, mobile robot motion control technology is facing the key problem of intelligent path planning. The combination of a grid map and an artificial intelligence algorithm provides an effective solution for path planning. However, the current grid map generation mainly relies on laser and ultrasonic sensors to obtain environmental information for grid map modeling, which is time-consuming and laborious. To address this issue, this paper investigates a method to transform a non-standard real environment map into a standard grid map. The non-standard real environment map was processed, and the size and shape of the map were standardized. The obstacles in the map were designed as a circle; the edge was expanded to add safety distance. The experimental results show that the method can transform real environment photos into standard grid maps suitable for robot path planning, and the path planning and motion control of robots in real environments can be carried out by using this method.

## 1. Introduction

In recent years, with the development of social productivity, mobile robot technology has also grown. Mobile robots, because of their ability to replace humans to complete a variety of work, have high productivity superiority, and are becoming the focus of research in various countries. Mobile robots can replace waiters in restaurants to pour water for customers [1,2], can replace workers in factory assembly lines to transfer all kinds of workpieces [3,4], replace humans in logistics warehouses to sort and store materials [5,6], can replace cleaners in families, hotels and public places to do some cleaning [7,8]. Mobile robots can also take the place of soldiers in front of the battlefield [9,10]. However, due to the disciplines involved in mobile robot technology, too many technologies, which involve a high intersection of technology, highly

**Data availability statement:** All relevant data are within the manuscript and its Supporting information files.

**Funding:** This research was funded by the Science and Technology Research Project of Henan Province, China (No. 222102210307). The funders had no role in study design, data collection and analysis, decision to publish, or preparation of the manuscript.

**Competing interests:** No authors have competing interests.

advanced mobile robot path planning technology as the main support to realize the function of mobile robots, are now still facing a very big challenge.

When the mobile robot completes its main work task, it not only needs to complete the special actions mentioned above, such as serving tea and pouring water, passing and cleaning, but also completes the movement from one position to another safely and smoothly. For mobile robots, to move from one position to another quickly and safely, it is necessary to plan and design the motion trajectory of the robot [11,12]. The initial position and the end position of the mobile robot are set, and then the path planning is carried out in the map by using various algorithms. According to the planned path, the robot is controlled to move according to the planned path to complete the movement of the mobile robot. Mobile robots can also use sensors, such as ultrasound, to continuously find a way to complete the movement from the initial position to the target position according to certain moving rules. However, on the one hand, this method has high requirements for sensors; on the other hand, it needs to constantly find a route, so it takes a long time and is not efficient [13–15].

In the research of robot path planning, Chuanxiang Ren et al. [16] improved the path planning of the artificial potential field method by using local information to solve the local optimality problem. Bo Fu et al. [17] proposed a method for improving pheromones, further improving the ant colony algorithm, and enhancing the algorithmic advantages of the ant colony algorithm. Peiying [18] used the A* algorithm to design a path planning algorithm for robot obstacle avoidance, which provided a reference for optimizing robot obstacle avoidance. Zengpeng Lu [19] introduced the adaptive learning and inversion strategy into the genetic algorithm, and improved the genetic algorithm path planning of the robot. However, whether it is ant colony algorithm path planning, artificial potential field method path planning, or genetic algorithm path planning, in addition to the algorithm itself, the robot motion environment map applying the algorithm should also be combined with it to achieve better and faster path planning [20,21].

The current robot movement environment map is mainly obtained by manual labor or by installing sensors such as lasers and ultrasonic waves on the robot. The peripheral information of the robot's activities is obtained through manual movement or the robot's movement. Establish the grid map of the robot's movement through this peripheral information. Or based on the known environmental information data, autonomously establish the grid map of the robot's movement [22,23] Zengzhen Mi [24] improved the real-time performance of map modeling and positioning by limiting the upper limit of the number of sampling particles; Longda Gao, Weiyang Lv, Xuyang Yan, and Yanzheng Han [25] pointed out in the article that environmental maps can be constructed in real time and covered by placing sensors on the robot. Tingchen Ma [26] obtained environmental information using 3D lidar for map modeling, which improved the modeling accuracy. Although these algorithms have improved the modeling methods, they all require a considerable amount of time or the use of high-precision sensors [27].

In today's world, artificial intelligence is developing very rapidly. As an advanced intelligent motion control system, Tesla's fully automatic driving FSD system can

realize the automatic driving of intelligent devices such as vehicles, but its operation mainly relies on Google Maps, and there is no motion control map in a small range. Motion control mainly relies on the camera for visual neural network control and uses ultrasonic sensors for automatic driving. If the camera can be used to obtain and automatically generate the motion control map in the real environment, it will not only greatly improve the intelligence of the system, but also greatly reduce the difficulty and cost of motion control.

In order to improve the efficiency of robot path planning and improve its intelligence degree, we studied the motion environment map modeling of a mobile robot in a real environment. We use mobile phones to take pictures of the robot motion environment, obtain the robot motion environment photos, and study the path planning method using the real environment photos of the robot motion. In addition, after the path planning of the robot, the use of the planned path for mobile robot motion control is also a key problem, which plays a key role in the motion control of the robot. In this paper, the standardization design process of non-standard real environment map is given, the obstacles in the real environment map are designed as circular and circular edge expansion, and the grid map of circular design of obstacles and circular edge expansion of obstacles in the real environment map is established. The path planning and pose planning of the ant colony algorithm under three grid maps of real environment grid map, obstacle circular design and obstacle circular edge expansion design grid map are studied. The method of using motion control code to control the motion of the robot is proposed, and the robot is used to carry out experimental verification. It opens up a new path for a wide range of robot motion control applications.

## 2. Methods

### 2.1. Overview of the main research steps

The main steps of this research are shown in Fig 1, which mainly include standard real environment grid map building, ACO path planning, and robot motion control according to the planning path. Firstly, the experiment obtains the photos of the real environment of the robot motion, and establishes the standard real environment grid map according to the real environment photos. Then, the path planning is carried out in the established standard real environment grid map, and the path and pose of the robot motion are planned. The robot motion control experiment is carried out to analyze and verify the model and algorithm proposed in this paper.

### 2.2. Non-standard real environment map standardized design

In order to better control the motion of the robot, it is necessary to plan the path of the robot. The path planning map generally refers to the map that can carry out the path planning algorithm [28,29]. In this paper, a non-standard real environment map refers to the photos of the robot motion environment directly obtained by a mobile phone. On one hand, the areas and environments shown in these photos are different. On the other hand, the photos themselves vary in size and shape due to being taken with different photographic equipment. Standardized design refers to processing the photos obtained by taking pictures with mobile phones. The process to obtain grid maps of the same size, shape and number of grids. These processes include scaling, rectangular box selection, cropping, binarization, griding, etc. The main purpose of the standardization design of a non-standard real environment map is to unify the processing of a non-standard real environment map, simplify the processing process of a real environment map, enhance the versatility of the method, and lay a good foundation for the intelligent standardization design of a non-standard real environment map.

Robot path planning in a variety of robot mobile environments usually needs to obtain the moving environment map of the robot first, and then carry out robot path planning according to the moving environment [30,31]. Therefore, before path planning, it is important to obtain a map of the robot's motion environment that can be applied to the path planning of the robot. Before path planning, we use the grid and binary method to process the moving environment map of the robot, and obtain the standard grid map for robot path planning.

**Fig 1. Main research steps.**

**2.2.1. Environment map space.** Consider a two-dimensional environment with known boundaries, where $W \subseteq R_2$ is a closed and bounded set representing all points in the environment. The coordinate point $W$ can be represented by two coordinates (x, y). In this two-dimensional environment, there are a large number of obstacles $Z_{obs}$, and we can know that $Z_{obs} \subseteq W$ in the colored area in the figure. Obstacles $Z_{obs}$ are defined as points in 2D space that cannot be reached. The other spaces are defined as spaces $Z_{move} \subseteq (W \backslash Z_{obs})$, and these spaces denote the spaces that can be reached. Within the space $W$, in the region $[L_x, L_y]$ to which $W$ belongs, we build a grid map.

**2.2.2. Environment map size.** In order to standardize the map processing, we need to unify the size of the map to be processed. We define the size of the region as $L_x \times L_y$.

$$\begin{cases} L_x = The \quad lateral \quad edge \quad length \\ L_y = The \quad vertical \quad edge \quad length \end{cases}$$

(1)

Within the spatial region $[L_x, L_y]$ to which W belongs, we build a grid map. Because the spatial region $[L_x, L_y]$ to which W belongs is obtained by using different cameras or different mobile phones, it is different. This is not conducive to the establishment of a grid map for motion path planning. Therefore, in order to standardize the map processing process, we want to unify the size of the map that needs to be processed. We define the size of the standard map in the processing as $L \times L$.

If

$$\text{Min}\,(L_x, L_y) > L \tag{2}$$

The photo is scaled down by a factor:

$$C = L/\text{Min}\,(L_x, L_y) \tag{3}$$

Then, we crop the image with the **imcrop** function with the cropping region $X \in [X_s, X_e]$ and $Y \in [Y_s, Y_e]$.
   Among them:

$$\begin{cases} X_s = (L_x/2) - (L/2) \\ X_e = (L_x/2) + (L/2) \\ \qquad Y_s = 0 \\ \qquad Y_e = L \end{cases} \tag{4}$$

Note that the path and calibration object we want to plan must be within the $L \times L$ interval.
   After determining the robot motion environment, we select the box of the robot motion environment map, and cut it to obtain the path planning area. We will carry out path planning for the robot in it.
   **2.2.3. Identify obstacles in the environment.** Generally, the photos obtained with mobile phones are color photos, which are processed in black and white to prepare for obstacle recognition. Then, we not only determine the obstacles, but also obtain the location and size information of each obstacle in the map through obstacle recognition. We use the **regionprops** function of MATLAB software to identify obstacles, get the length $L_{col}$ and width $L_{cow}$ of each obstacle, and surround the obstacles with rectangular boxes.
   The real size of the obstacle as reference in the upper right corner of the map is $L_X$ and $L_Y$, its size in the MATLAB system are $L'_{cow}$ and $L'_{col}$, we can know the scale of the whole figure.

$$BLC = L'_{Cow}/L_X = L'_{Col}/L_Y \tag{5}$$

The reference given in this study is circular. By the identification of reference, for which we take:

$$BLC = (L'_{Col} + L'_{Cow})/2D \tag{6}$$

$D$ is the radius of the circular reference.
   In order to accurately determine the real size of the map and the coordinates of the planning path in the later stage, reference objects are usually put into the real robot mobile environment, and the real coordinate position and planning path are determined through the reference.
   **2.2.4. Environment gridding.** In order to standardize the design of the map, we take the map from the top left corner as the starting point, horizontally to the right, and vertically down after grayscale. We divide the side length $L$ into $N$ parts. Then the number of grids in each row is $N$, the number of grids in each column is $N$, and the length of each grid is

$$h = L_y/N \tag{7}$$

In this way, a grid matrix consisting of multiple grids can be formed, where there is a total of $N \times N$ grids.
   In order to facilitate the ant colony algorithm path planning, we transform the grid matrix divided into the environment grid into a prescribed size $N \times N$.

$$\mathbf{B} = \text{imresize}\left(L, [N\ N]\right) \tag{8}$$

The number matrix is converted to the specified size, $\mathbf{B}$ is an $N \times N$ matrix, in which the maximum value is 255 (representing white) and the minimum value is 0 (representing black).

**2.2.5. Map binarization.** Because the general grid map needs to be built through the binary matrix, the binary matrix is also a key step in the standardization process. We can use the **floor** function to binarize the matrix $\mathbf{B}$.

$$\mathbf{J} = \textbf{floor}\left(\mathbf{B}/255\right) \tag{9}$$

We divide each element of $\mathbf{B}$ by 255; the number less than 255 divided by 255 is equal to 0 (black), the number equal to 255 divided by 255 is equal to 1 (white). Thus, the binary matrix composed of standardized 0,1 is generated.

**2.2.6. The establishment of a standard grid map.** For the original map a, the map length is set

$$\begin{cases} D_{\text{length}} = L \\ D_{\text{width}} = L \\ h = L/N \end{cases} \tag{10}$$

According to the above parameters, a grid map with side length $L$ and grid number $N \times N$ is drawn. The grid map is filled with 0 and 1, where the grid with 0 is filled with black, that is, obstacles. In order to show the expansion area of the map, we fill these areas with color, that is, the expanded area during the circular expansion design of the obstacle. The other area we fill them with white, that is the safe area.

**2.2.7. Grid coordinate.** This time, we take the area of the new $N \times N$ grid map as the limited motion area of path planning, denoted as SG. This area is called the standard grid map area. Take the top left corner of the map as the origin of the coordinates, the horizontal to the right is the positive direction of the X axis, and the vertical down is the positive direction of the Y axis. Here, we set up a rectangular coordinate system XOY in which the $N \times N$ grid map lies, within this limited motion region:

$$\begin{cases} X \in [0,L] \\ Y \in [0,L] \end{cases} \tag{11}$$

At the same time, since we have gridded the map into $N \times N$ grids, the $i$ th column we denote as $x_i$, and the $j$ th row we denote as $y_j$. In this way, the position of any grid in SG can be represented by $(x_i, y_j)$, which is the grid in column $i$ and row $j$.

If we take the position of the center of each grid as the position of the grid, according to the length h of each previous grid, we can deduce

$$\begin{cases} X_i = x_i \cdot h - 0.5 \cdot h \\ Y_j = y_j \cdot h - 0.5 \cdot h \end{cases} \tag{12}$$

**2.2.8. Grid coding.** In order to find the location of each grid on the map faster, we encode each grid. Starting from the origin of the upper-left coordinate of the SG region, the encoding method continues to give the encoding from left to right and from top to bottom. In this way, we assign an encoding value code to each grid. For the grid $x_i, y_j$

$$C_{\text{ode}} = \left(y_j - 1\right) * N + x_i \tag{13}$$

In this case, if we know the grid encoding value, we can get:

$$\begin{cases} x_i = \text{mod}(C_{ode}/N) \\ y_j = \text{int}(C_{ode}/N) + 1 \\ X_i = \text{mod}(C_{ode}/N) \times h - 0.5 \times h \\ Y_j = (\text{int}(C_{ode}/N) + 1) \times h - 0.5 \times h \end{cases} \tag{14}$$

If the coordinate system proceeds from left to right and bottom to top, then

$$\begin{cases} x_i = \text{mod}(C_{ode}/N) \\ y_j = \text{int}(C_{ode}/N) + 1 \\ X_i = \text{mod}(C_{ode}/N) \times h - 0.5 \times h \\ Y_j = N \times h - (\text{int}(C_{ode}/N) + 1) \times h + 0.5 \times h \end{cases} \tag{15}$$

Here, $\text{mod}(Q/N)$ denotes the remainder operation over $Q/N$, and $\text{int}(Q/N)$ denotes the integer operation over $Q/N$.

$x_i$ and $y_j$ denote the grid with the $Q_{th}$ grid in column $x_i$ and row $y_j$, and $X_i$ and $Y_j$ denote the center position of the $Q_{th}$ grid as $(X_i, Y_j)$.

## 2.3. Non-standard real environment standard grid map ant colony algorithm path planning process

After the standardized design of the non-standard real environment map, we obtain the standardized grid map, which can be used with the ant colony algorithm for path planning in the standardized grid map. Ant colony algorithm is a classical path planning algorithm, this article uses it for path planning [32–36].

We consider the initial position and target position of the robot. After obtaining the standardized grid map, we find the initial position code and the target position according to the position coding in the map.

For each path planning, after setting the initial position and goal position, we can set different parameter values according to the ant colony algorithm. In general, the parameters of the ant colony algorithm mainly include the number of ants $m$, the information heuristic factor $a$, and the pheromone evaporation factor $\rho$.

In the process of path planning, we input the relevant parameters, the start position code, and the end position code. The ant colony algorithm iterates continuously to find the optimal path, and then the planned path of the robot can be obtained. We assume that the set of points traversed by the planned path is $P_L = \{P_{LQ}|P_{LQ1}, P_{LQ2}, P_{LQ3} \dots P_{LQn}\}$

In $P_L$, $P_{LQ1}$ is the initial position, $P_{LQn}$ is the target position, and the

$$|P_L| = n \tag{16}$$

In $P_L$, $P_{LQ1}$ is the initial position, $P_{LQn}$ is the target position, and the path starts from the initial position, bypassing the obstacles and passing through the center position of each grid point in $P_L$ to reach the target position. In this way, the coordinates of each point are $(P_{LQxi}, P_{LQyi})$, so the set $P_{LQP}$ of coordinate positions passing through these points is

$$P_{LQP} = \left\{ (P_{LQPx1}, P_{LQPy1}), (P_{LQPx2}, P_{LQPy2}) \dots (P_{LQPxi}, P_{LQPyi}) \dots (P_{LQPxn}, P_{LQPyn}) \right\} \quad i \in [1, n-1] \tag{17}$$

Here, the coordinates $(P_{LQxi}, P_{LQyi})$ of these points can be obtained by Eq (14)

After these points, the path is divided into $n$-1 segments, and the direction of each segment represents the pose of the robot during this movement.

$$P_{LQAi} = (P_{LQPy(i+1)} - P_{LQPyi}) / (P_{LQPx(i+1)} - P_{LQPxi}) \quad i \in [1, n-1] \tag{18}$$

Through the above analysis, following the process of the ant colony algorithm to execute the program, we can obtain the planned path and pose of the robot.

## 2.4. Non-standard virtual environment map standard design simulation

In order to better study the standardized design of non-standard real environment map, we first establish a virtual environment map, as shown in Fig 2a. As can be seen from the figure, there are eight obstacles in Fig 2a, and these obstacles have different sizes, shapes, and colors. Because the size and shape of the environmental maps obtained through different methods are different, we first scale and select the rectangular box, then crop it.

The size of Fig 2a used in this study is 4160×3120, so $Lx=4160$, $Ly=3120$.

Then, we take $L=960$ as the side length of the standard map, as

$$\text{Min } (L_x, L_y) = 3120 > L$$

The photo is scaled down by the factor:

$$C = L/\text{Min } (L_x, L_y) = 960/3120$$

So

$$\begin{cases} L_x = 4160 \times 960/3120 = 1280 \\ L_y = 960 \end{cases}$$

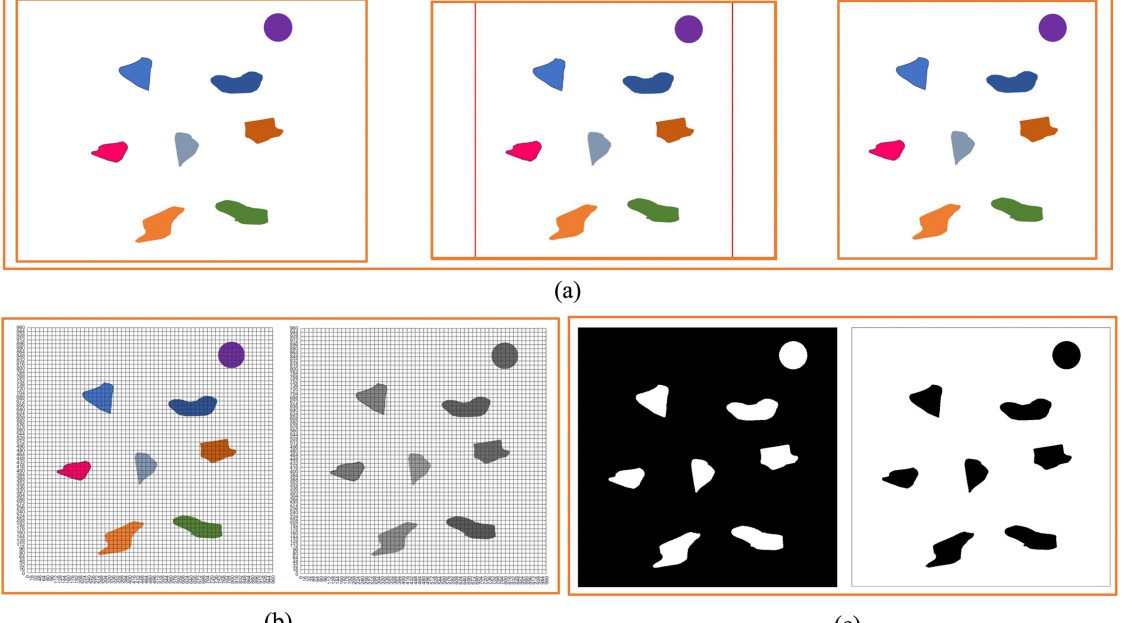

**Fig 2. Virtual environment map. (a)** Virtual environment map cropping process. **(b)** Virtual environment map grid division and grayscale processing. **(c)** Black and white virtual environment map.

And

$$\begin{cases} X_s = (L_x/2) - (L/2) = 160 \\ X_e = (L_x/2) + (L/2) = 1120 \\ Y_s = 0 \\ Y_e = L = 960 \end{cases}$$

The map is standardized in terms of size and shape through such a method. We perform black and white processing on Fig 2a to obtain Fig 2c. We can see that Fig 2c has two forms. Generally, through Fig 2c, we can identify the obstacles (Fig 3a) and obtain the size and location information of the obstacles. A total of 8 obstacles are identified in Fig 2a. The specific information of the identified obstacles in Fig 2 is shown in Table 1.

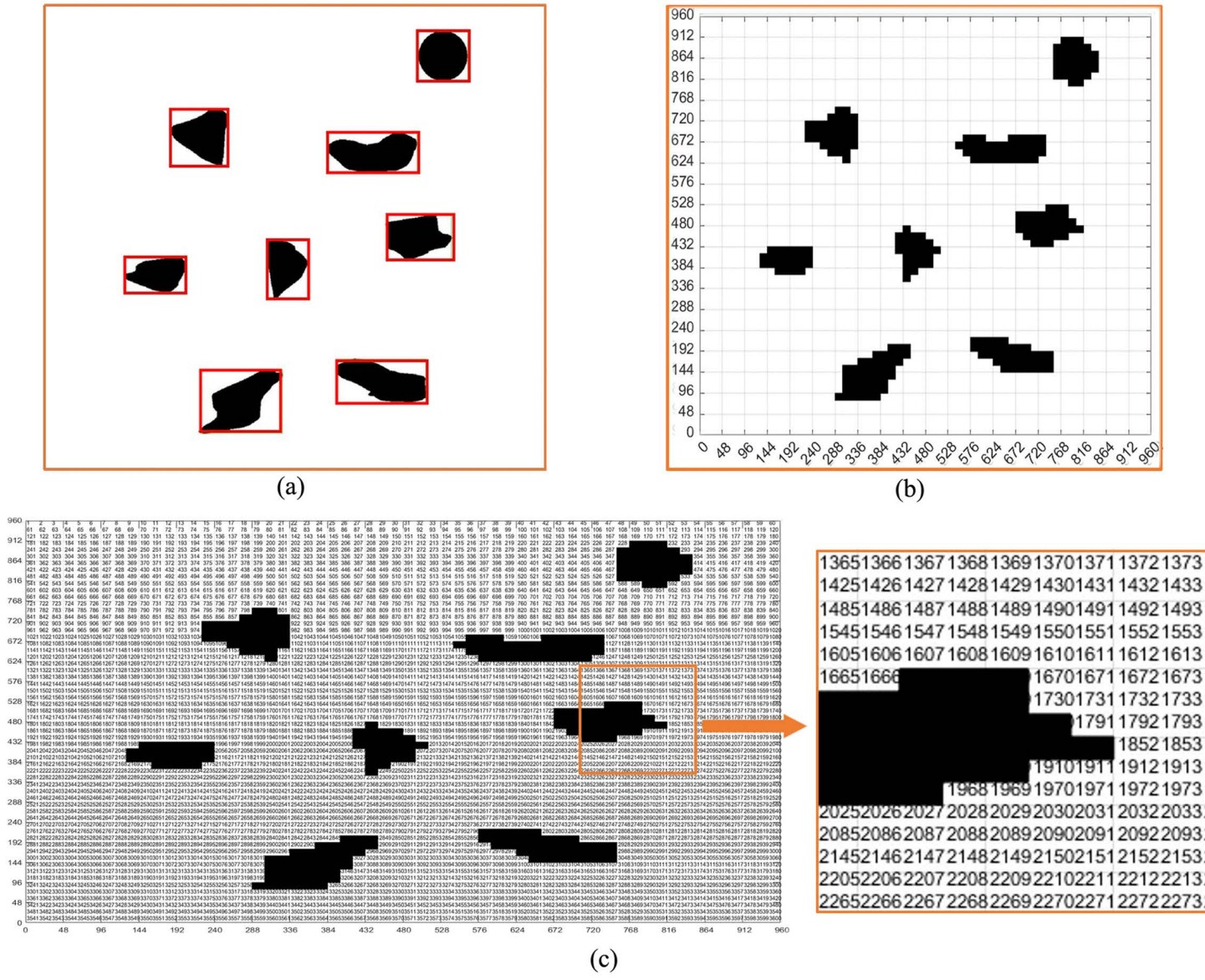

**Fig 3. Virtual environment grid map and coding. (a)** Obstacle box selection. **(b)** Binarized grid map. **(c)** Binarized grid map coding.

**Table 1. Obstacle information in the virtual environment map.**

| Number | x coordinate | y coordinate | Width | Length |
|---|---|---|---|---|
| 1 | 114.5 | 519.5 | 133 | 77 |
| 2 | 215.5 | 212.5 | 121 | 120 |
| 3 | 278.5 | 756.5 | 175 | 132 |
| 4 | 420.5 | 486.5 | 87 | 124 |
| 5 | 550.5 | 264.5 | 191 | 80 |
| 6 | 568.5 | 733.5 | 192 | 90 |
| 7 | 675.5 | 433.5 | 140 | 94 |
| 8 | 744.5 | 51.5 | 105 | 104 |

We grid Fig 2a to obtain Fig 2b. Through Fig 2b, and generate the standard grid map (Fig 3b). Fig 3b shows the grid map generated by the standardized design. The size of the map is set to 960×960, with a total of 60×60 grids, that is, 3600 grids, and each grid has an edge length of 16.

We define the position of each region in the map by coding, as shown in Fig 3c. The overall coding situation of the grid map can be seen from the figure, which is very important for us to provide the initial position, end position, position and pose setting for the path planning later. The encoding principle can be seen in Table 2.

It is because we define each position in the figure that our subsequent path planning can be completed successfully.

## 2.5. Virtual environment standard grid map path planning

Fig 4a-b shows the results of path planning in the original virtual environment based on the ant colony algorithm. We can see the planning path and pose in Fig 4. We expect that the planning path and pose can be fed back to the original environment map, and the scale BLC of the map is obtained according to the reference object. Through the scale, we feed back the planning path to the original map and get Fig 4c-d. The map clearly shows the planning path, the planning pose, and the position relationship between obstacles.

It can be seen from the figure that the planning path successfully bypasses the obstacles and reaches the goal. However, since the planning path is an optimized path, in order to minimize the movement displacement, the planning path is very close to the obstacles.

Due to the reasons of the robot's own structure, the planning path may lead to collisions between the robot and the obstacle if it does not fit with the obstacle. Therefore, if the distance ($L_{ij}$) between each point in the planning path and the obstacles

$$L_{ij} <= \text{Safe distance } \delta \tag{19}$$

**Table 2. Grid map encoding principle.**

| Algorithm MAP code |
|---|
| **Input:** Map length a, width b, and grid edge length c |
| Output: Map code |
| 1:**for** each column 1:a/c-1 |
| 2: **for** each line 1: b/c-1 |
| 3: d = num2str((i-1)*a/c + j); |
| 4: text(x,y,d); |

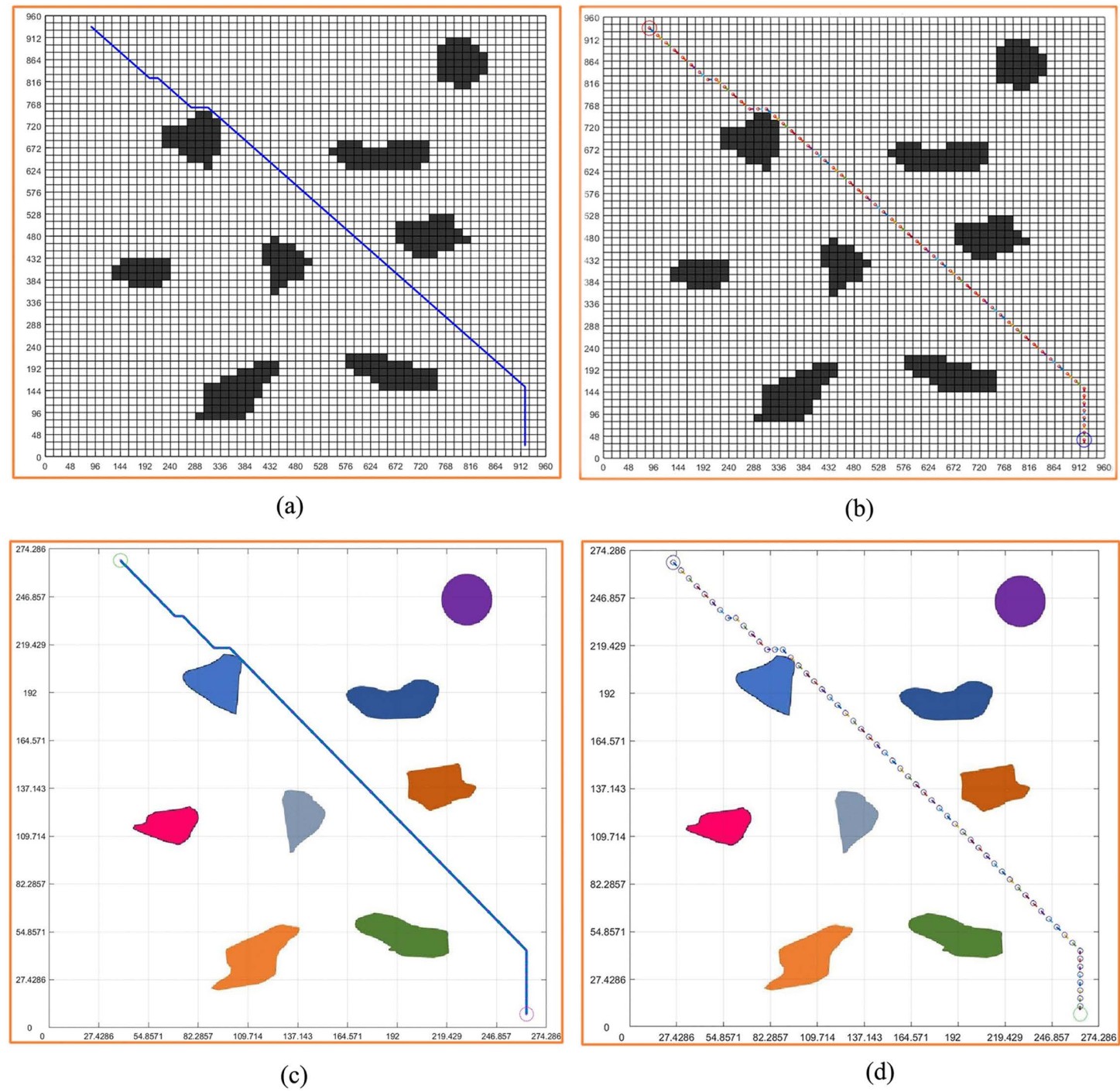

**Fig 4. Virtual environment grid map path planning. (a)** Virtual environment grid map planning path. **(b)** Virtual environment grid map planning pose. **(c)** Virtual environment grid map planning path and virtual environment integration. **(d)** Virtual environment grid map planning pose and virtual environment integration.

Then all of these points are unsafe. We take the set of all dangerous points $\mathbf{W} = \{w_1, w_2 \ldots w_j\}$.

In order to ensure that the robot can bypass the obstacles smoothly, we need to find new methods for path planning, so that the distance between each point in the planned path and the obstacles

$$L_{ij} > \text{Safe distance } \delta \tag{20}$$

Let's set the danger points

$$\mathbf{W} = \varnothing \tag{21}$$

Only in this way, it can be guaranteed that the planning path, pose can be applied to the robot motion control.

## 2.6. Virtual environment obstacle circular design and obstacle circular edge expansion design

In the complex robot mobile environment, the shape and size of each obstacle are different. In order to standardize the design of the real environment map, ensure the safety of the planning path, we design the obstacles in a circular way in the process of standard design of the environment. On the one hand, the standard design of the map can be completed, and on the other hand, the safety of the planning path can be guaranteed.

### 2.6.1. Obstacle circular design standard grid map.

Before the circular edge expansion for the obstacles, we first design the obstacles in a circular way. According to the map obstacle identification, we get the $L_{\text{Col}}$, $L_{\text{Cow}}$ of each obstacle, see Table 1, so we set

$$R_i = \left(L_{\text{Col}} + L_{\text{Cow}}\right)/2 \tag{22}$$

$R_i$ is the radius of the $i$th obstacle's circle. $L_{\text{Col}}$ and $L_{\text{Cow}}$ are the length and width of obstacle recognition respectively.

Subsequently, we design the obstacle in a circular and draw the circumcircle of the obstacle. According to Eq (22), each obstacle is designed in a circular way. The specific situation of the obstacles can be seen from Fig 5a, including the rectangular box and the outer circle of the reference object. Based on Fig 5a, the obstacles are covered to obtain Fig 5b. Then, the map was gridded (Fig 5c).

From Fig 5c, the standard grid map after the circular design of obstacles was obtained, that is, Fig 6. Comparing Fig 6 with Fig 3b, we can find that the two standard grid map have changed significantly. Each obstacle in the two figures is different. However, the obstacles in Fig 6 are more regular and appear circular, but with different sizes. From Fig 5a, we can see that although we design the obstacle in a circular way, some areas are still very close to the edge of the obstacle. For example, the bottommost obstacle in Fig 5a, whose lower left corner is in contact with the circle, may still collide with the planning path. Therefore, it is necessary for us to continue to design the obstacle in an extended way. According to Eq (22), we continue to add the safety distance $L_s$ to the radius of the circular design, then

$$R_{si} = \left(L_{\text{Col}} + L_{\text{Cow}}\right)/2 + L_s \tag{23}$$

$R_{si}$ is the safety radius of the $i$th obstacle's circular design. $L_s$ is safe distance. $L_{\text{Col}}$ and $L_{\text{Cow}}$ are the length and width of obstacle recognition respectively.

### 2.6.2. The standard grid map of the obstacle's circular edge expansion design.

As shown in Fig 7a, according to Eq (23), the inner circle is the circle of the circular design without safety distance, the outer circle is the circle of the circular edge expansion design with safety distance. The red part in Fig 7b is the part of the circular expansion design. At this time, continue to grayscale Fig 7b and carry out grid design to generate grid diagram Fig 7d. Next, Fig 8 is obtained according to the standard grid map generation method. After the obstacle circle design and the circle expansion design,

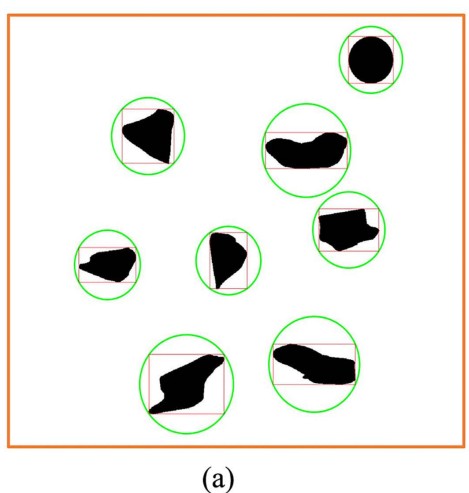 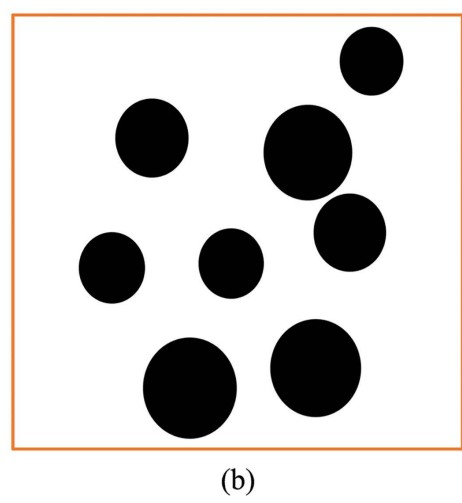 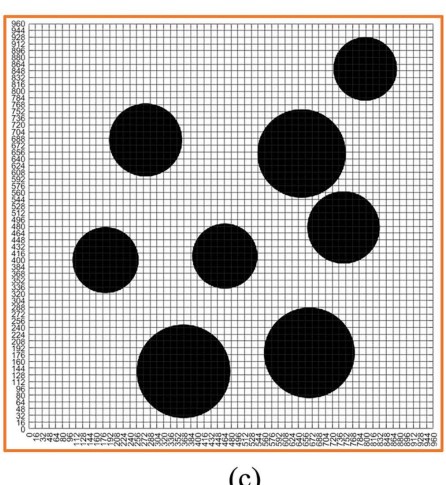

(a)                                 (b)                                 (c)

**Fig 5. Virtual environment obstacle circular design. (a)** Virtual environment obstacle circular circle selection result. **(b)** Circular coverage of obstacles in the virtual environment. **(c)** The obstacles of the virtual environment map are circled and the map is divided into grids.

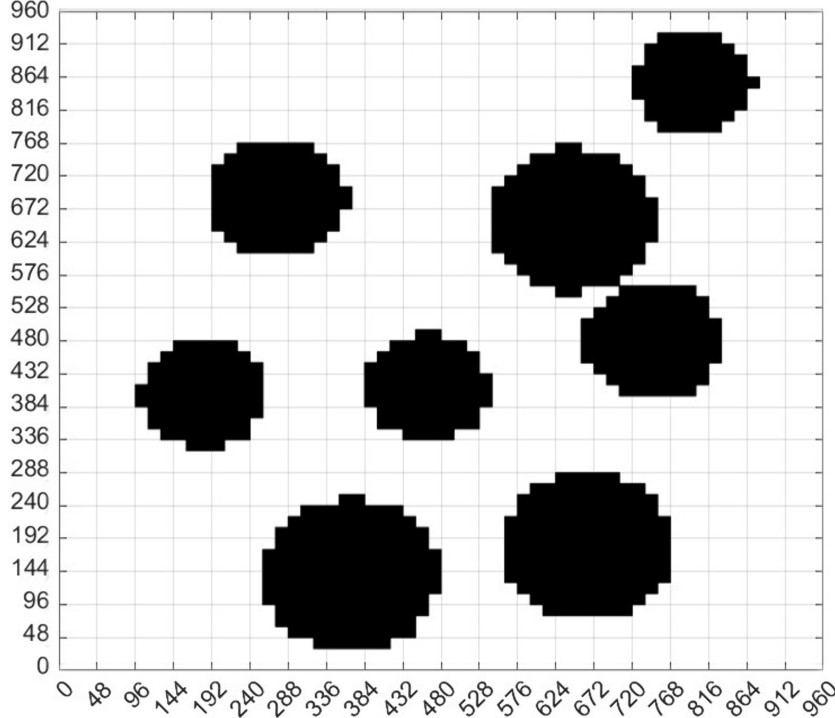

**Fig 6. Obstacle circular design map matrix and grid map.**

we can see that the obstacles in the standard grid map are circular, and the obstacles in the circular design are composed of grids one by one.

According to the geometric characteristics of obstacles, the obstacles in the non-standard environment are standardized and designed as circular figures composed of grids of the same size. These obstacles have been expanded on the

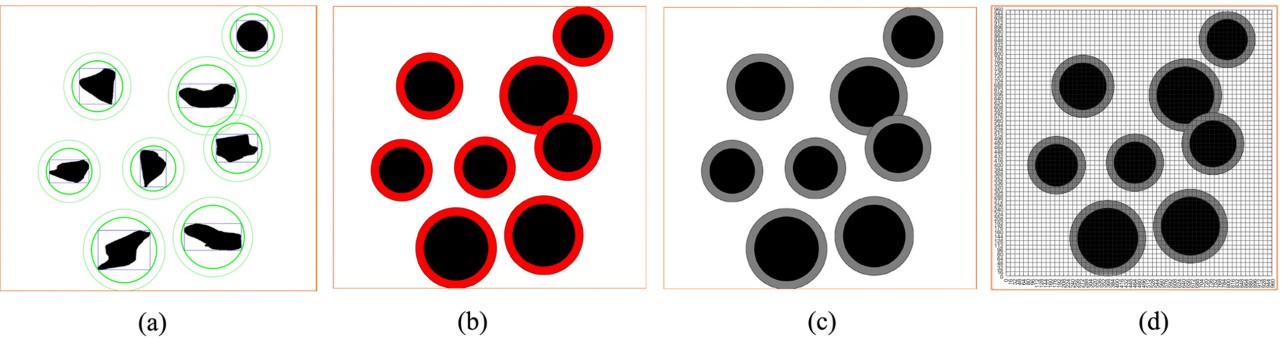

**Fig 7. The virtual environment obstacle circular edge expansion design. (a)** The virtual environment obstacles circular edge expansion circle selection. **(b)** Circular edge expansion circle coverage of obstacles in the virtual environment. **(c)** The virtual environment obstacle circle edge expansion design grayscale image. **(d)** The virtual environment obstacle circle edge expansion design grid graph.

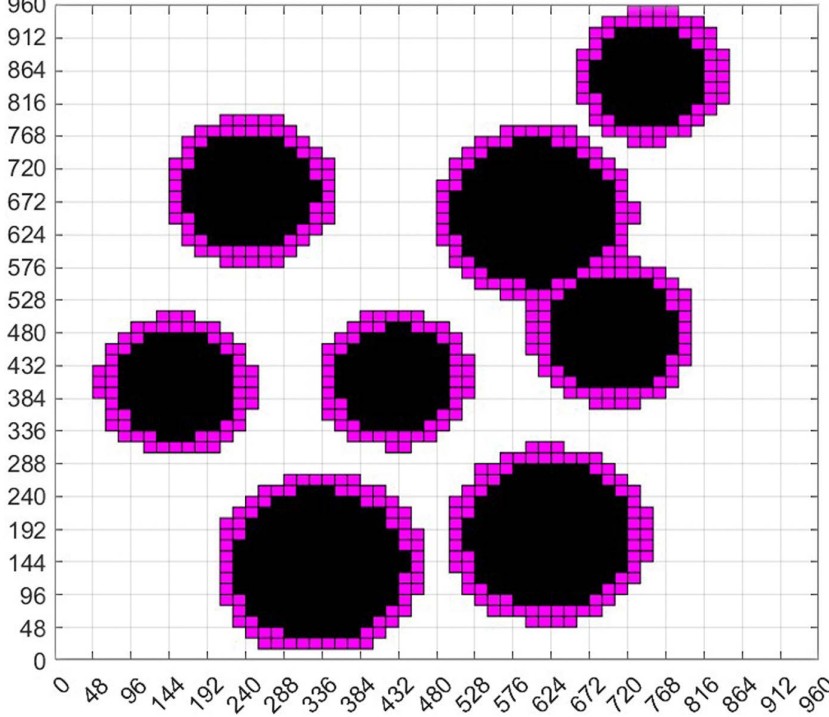

**Fig 8. Standard grid map for the expansion design of obstacle circular edges in virtual environment.**

basis of the geometric dimensions of obstacles in the original non-standard real environment map, and the safety distance has been added. The black area is the obstacle area, the purple-red area is the safe distance area, and the white area is the area that can move freely. In this way, we transform the real non-standard environment map into a standard grid map that can apply ant colony algorithm for path planning.

## 2.7. Virtual environment obstacle circular edge expansion grid map path planning

### 2.7.1. Standard grid map coding.
According to the grid coding method, we carry out coding design on the standard grid map, and the coding design results are shown in Fig 9. As can be seen from the figures, each part of the map gives the coding value. We select the initial position code and the end position code, the initial position and the target position are formulated, and then the path planning is carried out according to the ant colony algorithm.

### 2.7.2. Path planning of ant colony algorithm.
After coding, we set the initial position and goal position of the ACO path planning, and carry out path and pose planning according to Table 3.

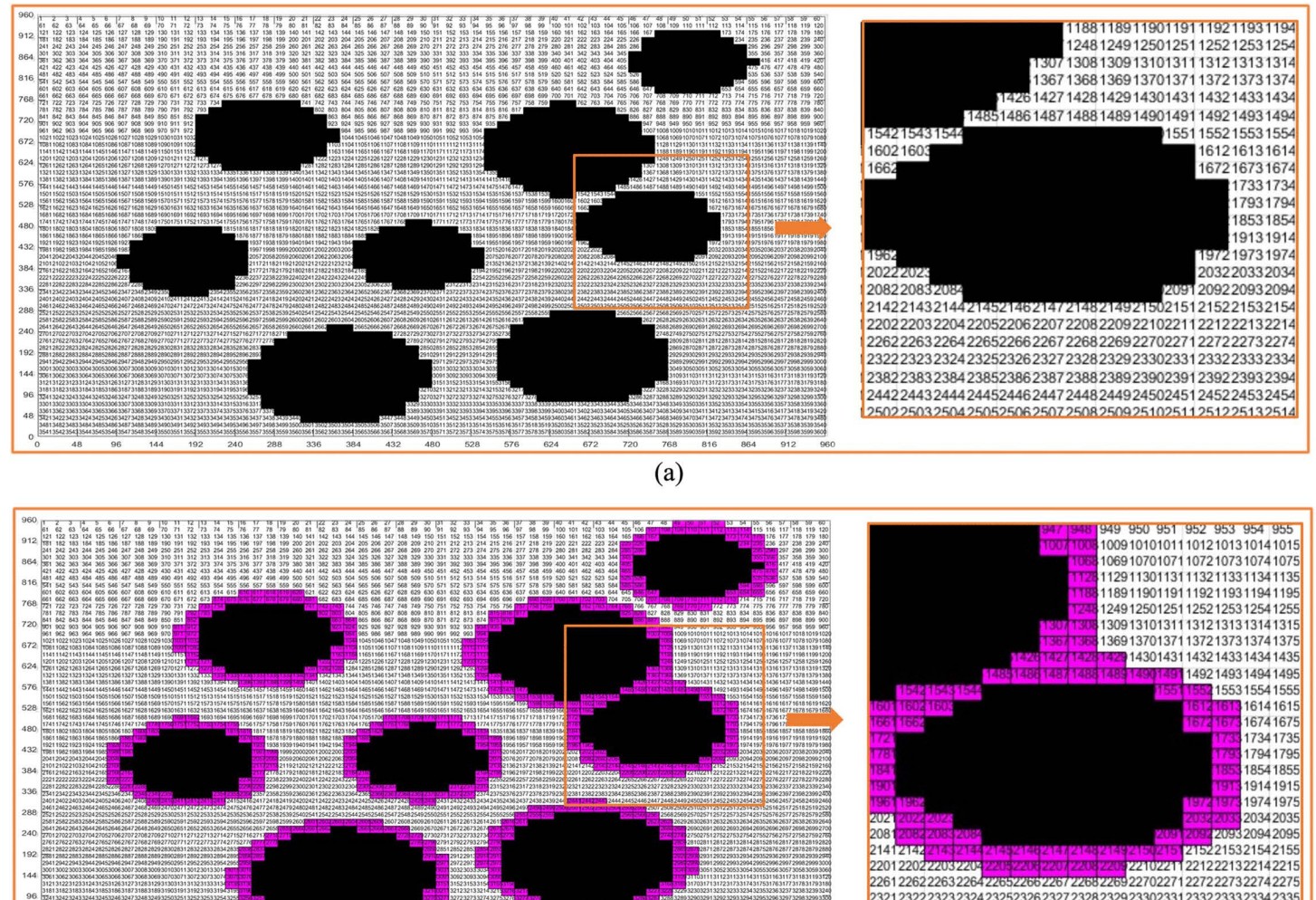

**Fig 9. Two method coding diagrams in virtual environment. (a)** Virtual environment obstacle circular design grid map coding diagram. **(b)** Virtual environment obstacle circular edge expansion design grid map coding diagram.

**Table 3. Parameters used for path planning of ant colony algorithm.**

| Parameters | Value | Parameter significance |
|---|---|---|
| K | 100 | Iteration number |
| M | 100 | Ant number |
| S | 66 | Encoding of starting position |
| E | 3538 | Encoding of target location |
| Alpha | 1 | Important parameter of pheromone |
| Beta | 30 | Parameters of heuristic factor importance |
| Rho | 0.2 | Pheromone evaporation coefficient |
| Q | 100 | The coefficient of increasing the intensity of pheromones |

Figs 10 and 11 are the paths planned by the standard grid map of circular design and circular edge expansion design. The standard grid map and the environment map are converted to each other by the reference scale, as shown in two figures.

Among the two figures, Fig 11 carries out circular edge expansion design due to the addition of safety distance, so the obstacles are relatively large. The planning path in the map is shown in the non-standard real environment map by passing the reference scale, as shown in Figs 10c and 11c. Through Figs 10c and 11c, we can see that the planning path in Fig 11c successfully bypassed the obstacle and reached the goal because of the addition of the safety distance, and the planning path in Fig 11c is safer than Fig 10c.

Figs 10d and 11d are the results of feeding the planned pose back into the virtual real environment map. By checking, we can know that the 17 points of the planned pose are dangerous, and we can change the position of the 17 points later to ensure the safety of the planned path. At the same time, due to the addition of safe distance, the planning pose in Fig 11d is safe throughout the path.

Fig 12 shows the path iteration diagram of the circular obstacle design and the circular obstacle edge expansion design. It can be seen from the figure that the path of the circular obstacle edge expansion design is longer than that of the circular obstacle design, which is due to the expansion of the obstacle design, which takes more paths than the circular obstacle design. Although some more paths are taken, it is these paths that ensure the safe movement of the robot.

From the above processes, it can be seen that the proposed circular edge expansion design of obstacles can realize path planning in the virtual environment, and theoretically this method can plan a safe path for robot.

## 3. Experiment and results

### 3.1. Real environment path planning experiments

After introducing the method theoretically, in order to verify the proposed design of obstacle circular edge expansion, the proposed theoretical method is verified by experiments. Firstly, the path planning experiment is carried out through the real non-standard environment map to verify the application of the method in the real environment map. Then the robot motion control experiment is carried out, and the path planning is carried out in the real environment map, the planning path and pose are used to control the robot motion, which verifies the effectiveness of the method proposed in this paper.

**3.1.1. Experiment non-standard real environment.** Before the path planning of the robot, we need to take photos of the environment where the robot is. Fig 13a shows the non-standard real environment. When doing this experiment, it is necessary to ensure that the map taken is clear and the photography area is within the path planning range of the robot. If there are obstacles with known size in the real environment map, the scale can be obtained by taking the obstacles with known size as the reference object. If there are no obstacles of known dimensions in the figure, a reference of known

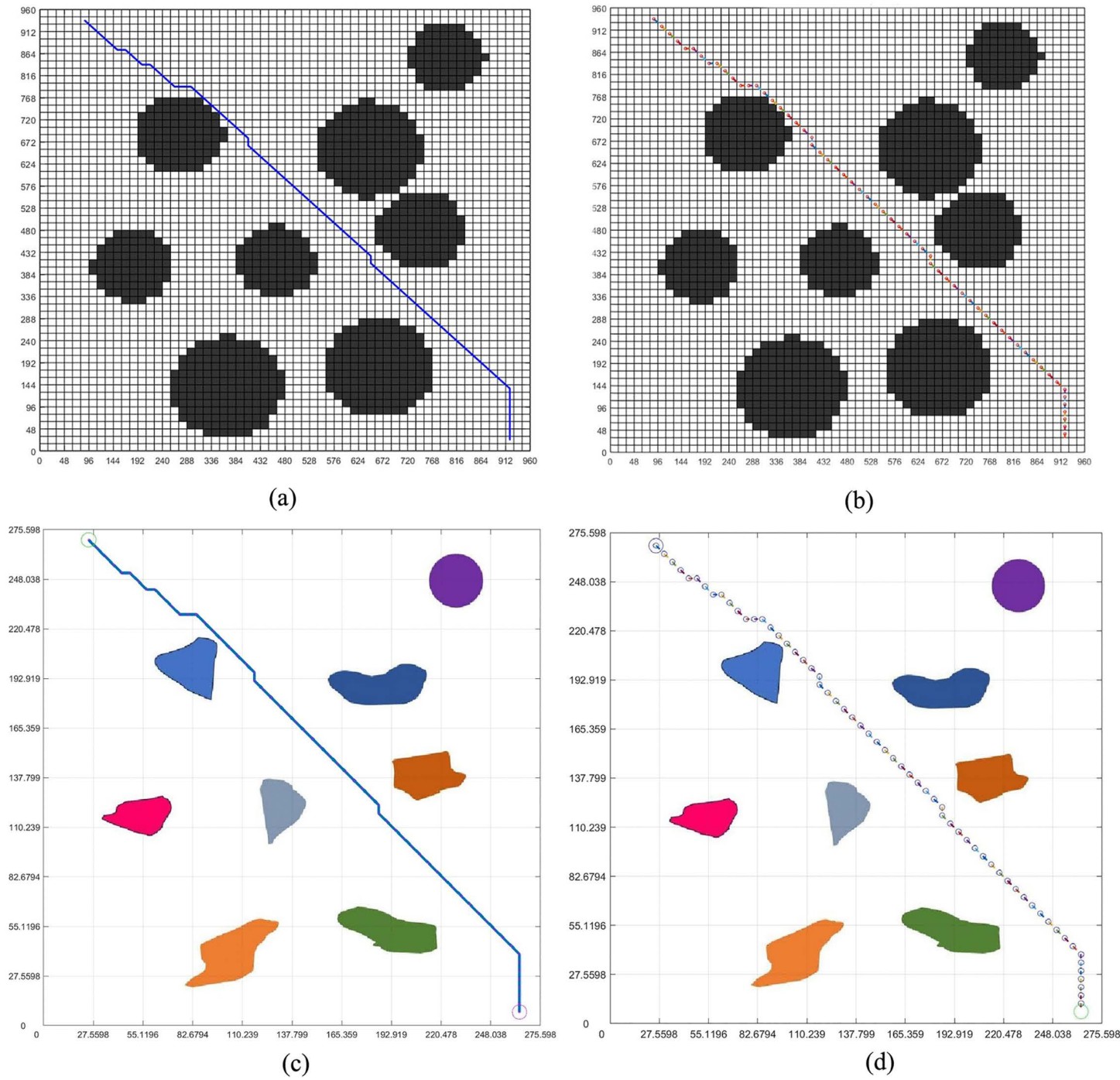

**Fig 10. Obstacle circular design path planning. (a)** Obstacle circular design grid map planning path. **(b)** Obstacle circular design grid map planning pose. **(c)** The integration of the grid map planning path and the virtual environment map. **(d)** The integration of the grid map planning pose and the virtual environment map.

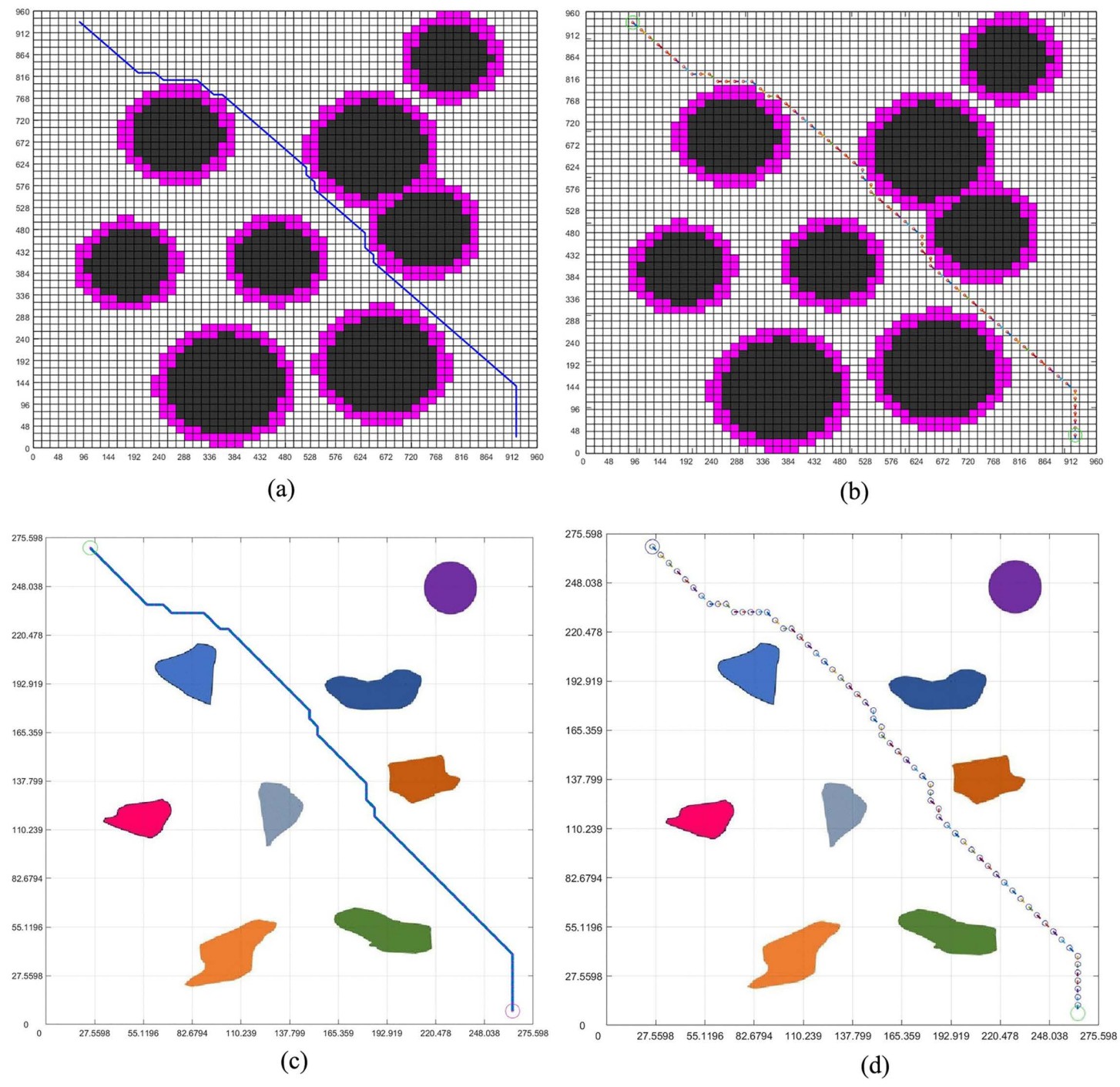

**Fig 11. Obstacle circular edge expansion path planning. (a)** Obstacle circular edge expansion grid map planning path. **(b)** Obstacle circular edge expansion grid map planning pose. **(c)** The integration of the grid map planning path and the virtual environment map. **(d)** The integration of the grid map planning pose and the virtual environment map.

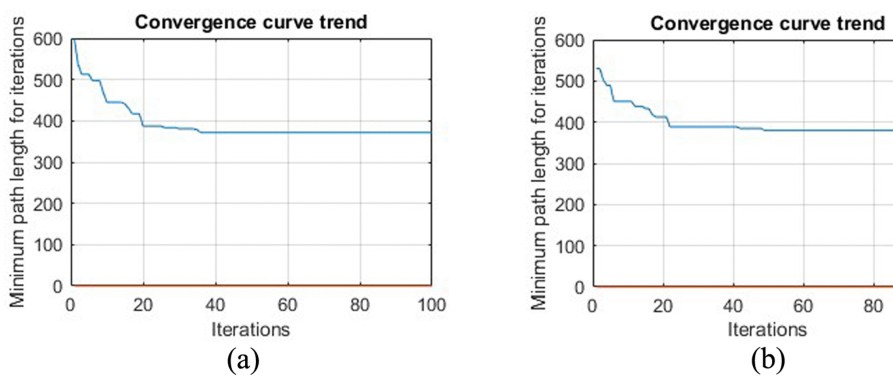

**Fig 12. Two method path planning iteration diagram in virtual environment. (a)** Virtual environment obstacle circular edge design grid map path planning iteration diagram. **(b)** Virtual environment obstacle circular edge expansion design grid map path planning iteration diagram.

(a)

(b)

(c)

**Fig 13. Real environment map. (a)** Real environment map cropping process. **(b)** Black and white real environment map. **(c)** Real environment map grid graph.

dimensions is placed in the environment, such as the black circular reference in the top right corner of the Fig 13a. The scale is obtained by means of a reference whose dimensions are already known as 200 mm.

**3.1.2. Non-standard real environment standard grid map generation experiment results.** Fig 13a was mapped according to the method introduced before. It underwent cropping, obstacle identification, and gridding. Here, the black and white processing of the original map Fig 13a can obtain Fig 13b. We generally use the right picture of Fig 13b as the basis for obstacle recognition to obtain obstacle size and location information in Table 4.

In Fig 13b, there are 7 obstacles of different sizes and 1 reference object that have been put into the environment. The size and position information of the reference objects are shown in Table 4 after identification. Fig 13a is divided into grids to obtain Fig 13c, and finally Fig 14 is obtained.

**Table 4. Obstacle information in the real environment map.**

| Number | x coordinate | y coordinate | Width | Length |
|---|---|---|---|---|
| 1 | 0.5 | 388.5 | 102 | 105 |
| 2 | 233.5 | 642.5 | 59 | 57 |
| 3 | 333.5 | 41.5 | 49 | 53 |
| 4 | 412.5 | 422.5 | 141 | 142 |
| 5 | 492.5 | 834.5 | 92 | 88 |
| 6 | 647.5 | 159.5 | 67 | 81 |
| 7 | 770.5 | 18.5 | 148 | 149 |
| 8 | 856.5 | 387.5 | 95 | 62 |

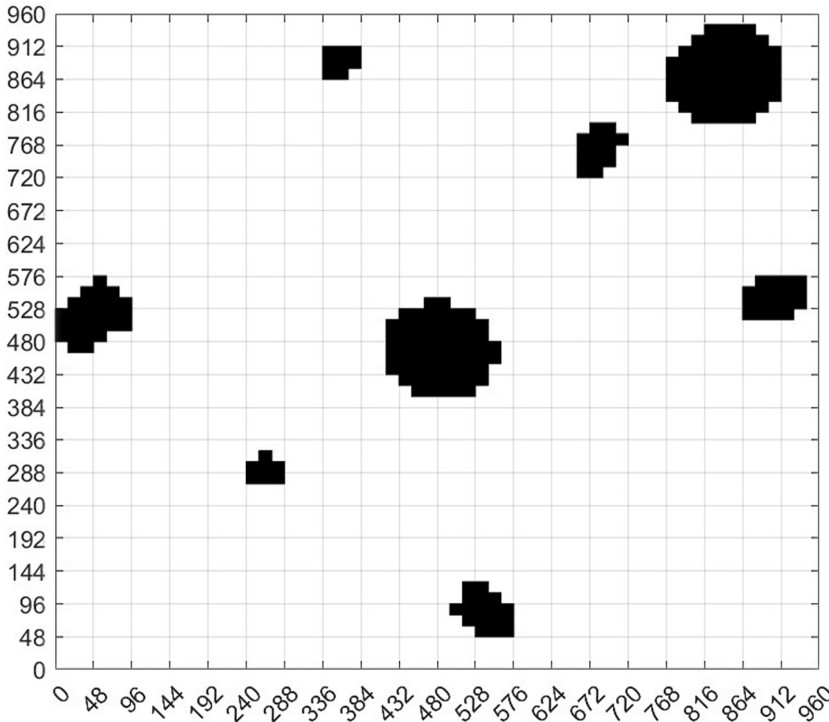

**Fig 14. Real environment grid map.**

**3.1.3. Real environment obstacle circular design grid map generation experiment results.** Fig 15 shows the process of circular design of obstacles in the real environment map. In the figure, according to the results of obstacle recognition, we circle the obstacle (Fig 15a). After circle selection, we fill the inside of the obstacle circle to obtain Fig 15b. Then, we divide the Fig 15b into grids (Fig 15c). The standard grid map generation is carried out through Fig 15c, thus establishing the standard grid map Fig 16. Comparing Fig 16 with Fig 14, we can see that we successfully obtained the standard grid map of obstacle circular design in non-standard real environment.

**3.1.4. Real environment obstacle circular edge expansion design grid map generation experiment results.** Fig 17 takes the circular design of obstacles in Fig 15 a step further. On the basis of the original circular design of obstacles, the edge of the obstacle circle of the circular design of obstacles is expanded, as shown in Fig 17a. According to the process introduced in the circular edge expansion design of obstacles, the real environment is processed, and Fig 17a-17c were obtained respectively. The obstacles in the original non-standard real environment map were circled and the circle edge was expanded. The grid map of the expansion design of the obstacle circular edge in Fig 18 is obtained according to Fig 17c. In the experiment, the average time taken from the original map to the map that can be used for path planning is about 20.87364 seconds. Just take a photo with your phone to obtain a grid map for robot motion planning and control, which reduces the difficulty of robot motion control and improves the intelligence level of robot motion control.

In order to better find the position in the graph, we also encode the real environment standard grid map and the real environment map as shown in Fig 19. In this way, we can quickly find the position of each grid in the map, which is convenient for subsequent path planning. It should be pointed out that the locations indicated by the points with the same serial numbers in the four coding maps are the same position. Therefore, the four maps are in one-to-one correspondence.

**3.1.5. Experiment results of path and pose planning under three maps.** Although the standard grid map of the non-standard real environment map, the standard grid map of the circular design of obstacles and the standard grid map of the circular edge expansion design are obtained, it is still necessary to carry out path planning and pose planning on the basis of these standard grid maps.

With the same starting position and the same target position, according to the maps generated by the three methods, we carried out the path and pose planning experiment. It can be seen in Fig 20a, the shapes of the obstacles are various.

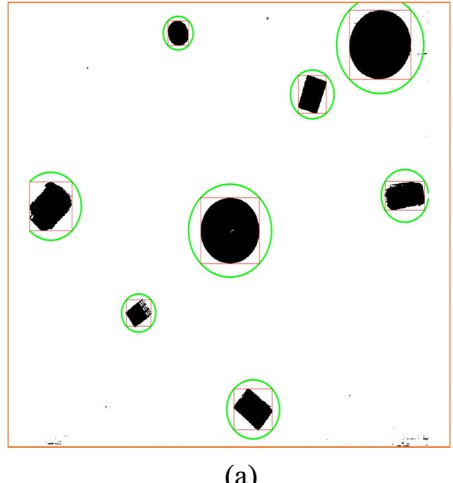 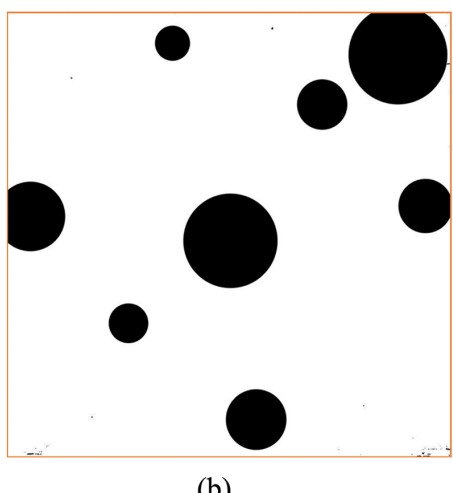 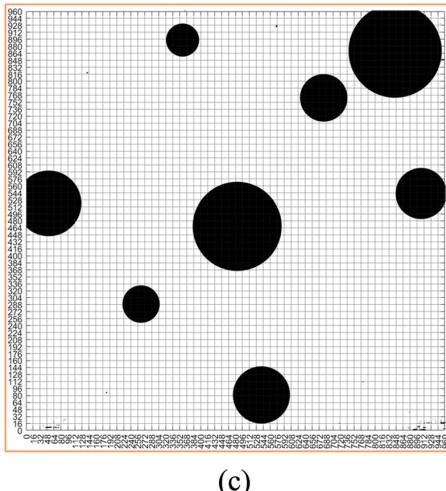

(a) (b) (c)

**Fig 15. Real environment obstacle circular design. (a)** Real environment obstacle circular circle selection result. **(b)** Circular coverage of obstacles in the real environment. **(c)** Real environment obstacle circular design map gridding.

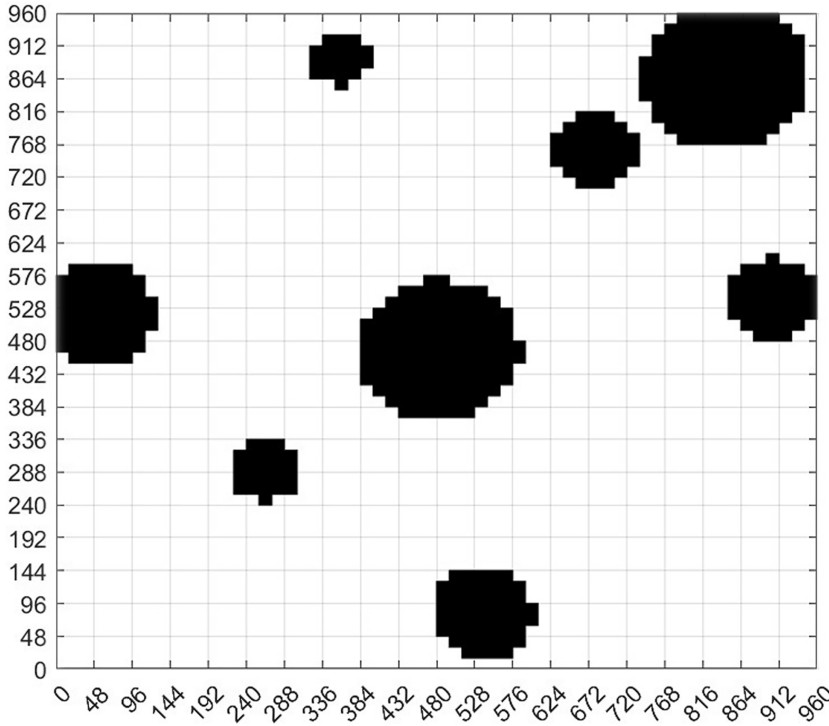

**Fig 16.  Real environment obstacle circular design grid map.**

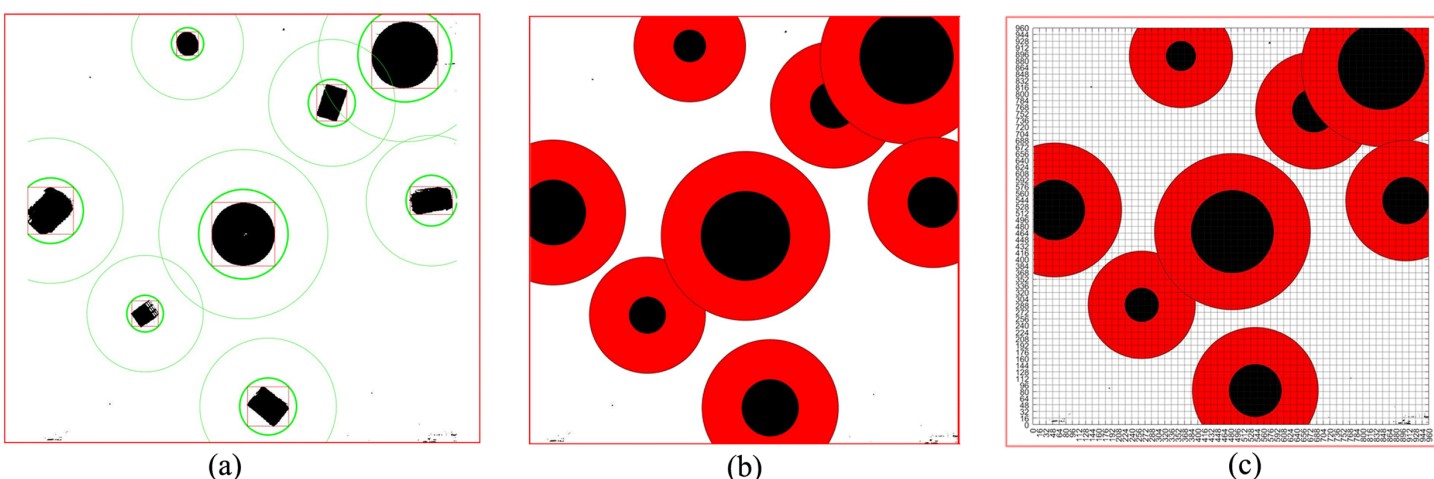

**Fig 17.  Real environment obstacle circular edge expansion design. (a)** Real environment obstacles circular edge expansion selection result. **(b)** Real environment obstacles circular edge expansion coverage. **(c)** Real environment obstacle circular edge expansion map gridding.

The shapes of the obstacles in Fig 20b and Fig 20c are more similar. By feeding back the planning path and pose in the grid map in Fig 20 to the real environment map, we obtain the planning path and pose of the real environment.

Fig 20a shows the path planning and pose planning results generated based on the non-standard real environment map. It can be seen from the figure that the planning path and planning pose from the initial position to the goal position

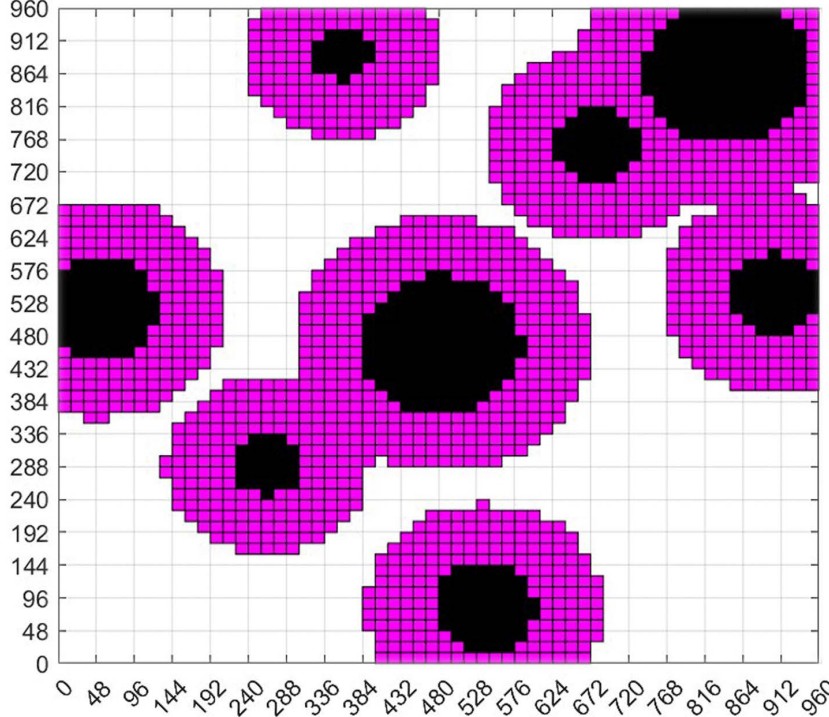

**Fig 18. Standard grid map for the expansion design of obstacle circular edges in real environment.**

contact with the obstacle in the center of the figure, and the path and pose are close to the obstacle to reach the goal position. This is because, as a path optimization algorithm, the generated path takes the minimum value of the iteration, so the planned path also chooses the shortest distance to continue walking to reach the target position when encountering obstacles. It will walk close to the obstacles when encountering them, and then bypass the obstacles.

Compared with the standard grid map with circular design of obstacles in Fig 20a, the planning path and pose in Fig 20b maintain a certain distance from the obstacles due to the circular design of obstacles. However, compared with Fig 20c with safety distance added, there are still some areas close to the obstacles. Although the obstacles are designed in a circular way, the safety distance is not added as a whole, and some areas are still in contact with the planned path. In order to ensure the overall safety of the planning path, we carry out a circular edge expansion design for the obstacles and add the safety distance. In Fig 20c, we can see that the overall planning path and pose maintain a certain distance from the obstacles, and the whole path and pose are safe. Compared with Fig 20a-Fig 20C, it can be seen that the overall planning path is far away from the obstacles.

The experiment system used in this paper is Windows 10 Professional Edition, and the MATLAB software was utilized. The CPU used is Intel(R) Core(TM) i5-10400F, with a base frequency of 2.90GHz. According to Table 5, the average values and standard deviations of the running time decomposition under three maps can be obtained, as shown in Table 6. From Table 6, the running time of each stage can be obtained. It can be seen from Table 6 that in the path planning of obstacle circular edge expansion design, the average time required to go from the original map to generating a grid map that can be used for path planning is 20.87364 seconds.

Fig 21 shows the analysis of the safety of the paths in three scenarios. Based on the size of the robot, we take the center of the robot as the point of the planned path and set the safe distance $\delta = 120\,\text{mm}$. In the figure, the orange-red grids represent the obstacles, the green ones represent the grids expanded after circularly designing the obstacles, and

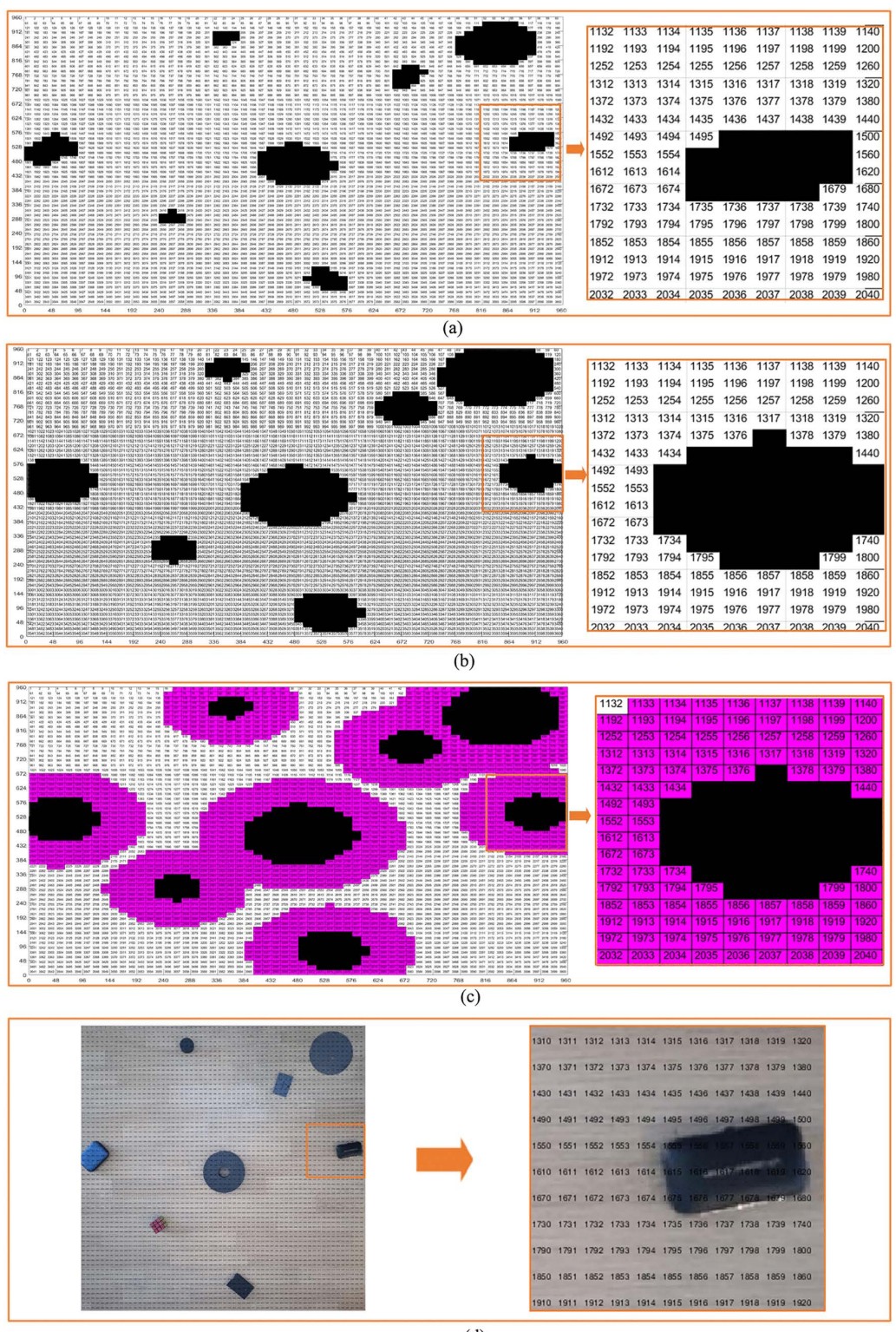

**Fig 19. Real environment grid map coding.** (a) Real environment original grid map encoding. (b) Real environment obstacle circular design grid map encoding. (c) Real environment obstacle circular edge expansion design grid map encoding. (d) Real environment map encoding.

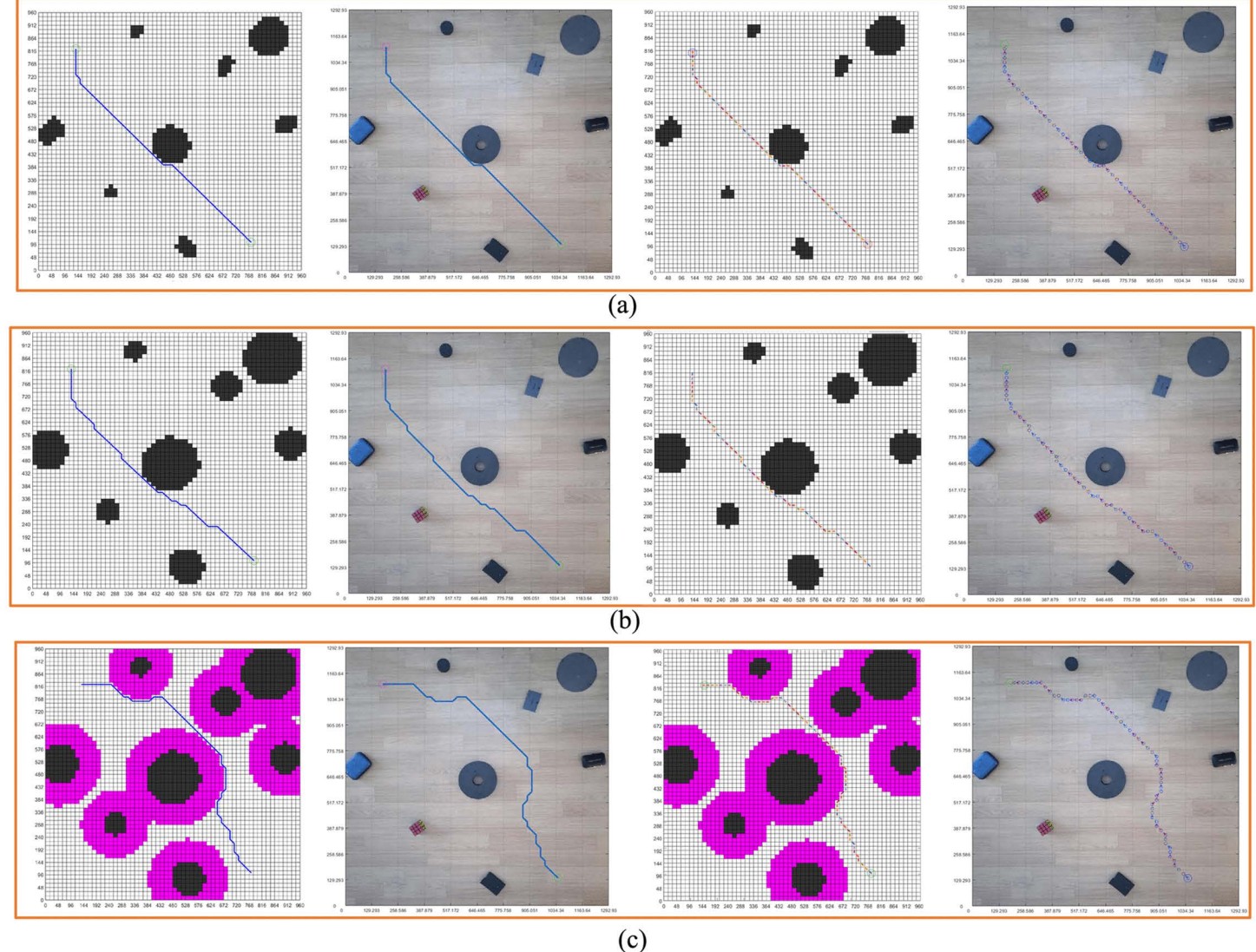

**Fig 20. Real environment three grid maps path planning. (a)** Real environment grid map planning path, pose, and real environment map display results. **(b)** Circular design of real environment obstacles grid map planning path, pose, and real environment map display results. **(c)** Real environment obstacle circular edge expansion design grid map planning path, pose, real environment map display results.

the purple-red ones represent the grids with the safety distance added after circularly designing the obstacles. The blank area outside the purple-red color in the figure is the safe area. The points where the paths in the orange-red, green, and purple-red grids are passed through in the planned path are dangerous path points, and the path points not in these three colors are safe points. Fig 21 clearly illustrates the safety of the planned paths. The planned path points in the three scenarios were displayed in the grid map and the real environment map. The circular path points in the figure are the safe planned path points, and the red rhombus-shaped planned path points are dangerous points. By dividing the number of safety points by the total number of path points, we can obtain the safety level of the planned path.

**Table 5. Real environment three maps path planning time decomposition.**

| Map | Runs number | Obstacle detection time | Circular design time | Edge expansion time | Binaryization time | Grid Time | Encoding time | ACO Iterations | ACO time |
|---|---|---|---|---|---|---|---|---|---|
| Real environment grid map | 1 | 1.868184 | no | No | 0.735870 | 0.908040 | 8.484127 | 100 | 517.917291 |
| | 2 | 1.780151 | no | No | 0.510136 | 0.854324 | 8.597538 | 100 | 518.929980 |
| | 3 | 2.226469 | no | No | 0.507457 | 0.847581 | 8.504462 | 100 | 519.234692 |
| | 4 | 1.794207 | no | No | 0.509332 | 0.859197 | 8.915158 | 100 | 518.444590 |
| | 5 | 1.935081 | no | No | 0.510216 | 0.870388 | 8.631679 | 100 | 528.664426 |
| | 6 | 1.872564 | no | No | 0.520992 | 0.846994 | 8.580158 | 100 | 527.404944 |
| | 7 | 2.036303 | no | No | 0.509820 | 0.844056 | 8.605856 | 100 | 524.992409 |
| | 8 | 2.034385 | no | No | 0.506885 | 0.877770 | 8.492951 | 100 | 527.218208 |
| | 9 | 2.808666 | no | No | 0.721007 | 1.133878 | 8.781284 | 100 | 532.284944 |
| | 10 | 1.931701 | no | No | 0.511859 | 0.853417 | 8.576773 | 100 | 529.597628 |
| Obstacle circular design grid map | 1 | 1.207452 | 0.909182 | No | 0.467535 | 1.372664 | 8.624966 | 100 | 521.119206 |
| | 2 | 1.799200 | .192494 | No | 0.515596 | 1.427930 | 8.829550 | 100 | 521.249879 |
| | 3 | 1.788790 | 1.203014 | No | 0.517731 | 1.441125 | 9.779416 | 100 | 526.769700 |
| | 4 | 1.823641 | 1.193224 | No | 0.513123 | 1.465887 | 8.820949 | 100 | 519.067217 |
| | 5 | 1.959911 | 1.174551 | No | 0.813992 | 1.514541 | 8.700301 | 100 | 519.456291 |
| | 6 | 1.783259 | 1.239933 | No | 0.537293 | 1.426484 | 8.702214 | 100 | 519.654425 |
| | 7 | 1.777295 | 1.204239 | No | 0.506757 | 1.440169 | 8.718007 | 100 | 519.383686 |
| | 8 | 1.980916 | 1.191190 | No | 0.750520 | 1.543056 | 8.695686 | 100 | 519.055389 |
| | 9 | 1.781141 | 1.197368 | No | 0.513500 | 1.439194 | 8.686779 | 100 | 519.293535 |
| | 10 | 1.954955 | 1.189383 | No | 0.506506 | 1.427802 | 8.772505 | 100 | 519.474263 |
| Obstacle circular edge expansion grid map | 1 | 2.002939 | 1.809842 | 0.924207 | 0.246698 | 4.879568 | 11.476017 | 100 | 528.136388 |
| | 2 | 1.785878 | 1.230086 | 0.930101 | 0.261311 | 4.899112 | 11.414426 | 100 | 501.335903 |
| | 3 | 1.785956 | 1.185113 | 1.009968 | 0.267032 | 4.967463 | 11.537013 | 100 | 504.677117 |
| | 4 | 1.824811 | 1.208638 | 0.946767 | 0.257466 | 4.916922 | 11.488857 | 100 | 487.371966 |
| | 5 | 1.803652 | 1.190683 | 0.937713 | 0.257000 | 4.927117 | 11.436593 | 100 | 506.396266 |
| | 6 | 1.824745 | 1.174658 | 0.964570 | 0.241339 | 4.940801 | 11.511532 | 100 | 489.119603 |
| | 7 | 1.838175 | 1.192041 | 0.944997 | 0.257288 | 4.938923 | 11.602851 | 100 | 487.041228 |
| | 8 | 1.778068 | 1.195138 | 0.965473 | 0.262098 | 6.075267 | 11.717248 | 100 | 508.147619 |
| | 9 | 1.795510 | 1.200404 | 0.960805 | 0.269303 | 4.987781 | 11.511487 | 100 | 493.281972 |
| | 10 | 1.788817 | 1.213075 | 0.970038 | 0.264738 | 4.996139 | 11.544158 | 100 | 489.414590 |

## 3.2. Robot motion control experiments

**3.2.1. Experiment system and equipment.** A mobile phone is placed above the experiment place, and the real environment photos of the robot movement are obtained through the mobile phone. The mobile phone used in this experiment is REDMI9A. Below the mobile phone is the environment where the robot moves. In the environment where the robot moves, different obstacles are placed, and these obstacles have different sizes and shapes. Fig 22e shows the robot used in the experiment, which is a ROBOROBO robot, its size is 180 mm × 180 mm. It consists of four steering wheels, each of which can move independently, and the overall movement is flexible. This robot can move in 8 directions. We assign the encoding values of 1–8 to the robot's pose during movement, and encode them based on the poses in the grid map or the real environment map, as shown in Fig 23.

According to the path and pose generated by the planning, the motion control of the robot is carried out to control the robot safely from the initial position to the goal position by using the planning path and pose, which is the main goal of this

**Table 6. Real environment three maps planning time decomposition mean and standard deviation.**

| Map | Item | Obstacle detection time | Circular design time | Edge expansion time | Binaryization time | Grid Time | Encoding time | Total time for generating grid map | ACO time |
|---|---|---|---|---|---|---|---|---|---|
| Real environment grid map | Average value | 2.028771 | No | No | 0.554357 | 0.889565 | 8.616999 | 12.08969 | 524.4689 |
| | Standard deviation | 0.304162 | No | No | 0.091898 | 0.087973 | 0.135519 | No | 5.361824 |
| Obstacle circular design grid map | Average value | 1.785656 | 1.069458 | No | 0.564255 | 1.449885 | 8.833037 | 13.70229 | 520.4524 |
| | Standard deviation | 0.219394 | 0.321756 | No | 0.117146 | 0.04817 | 0.338456 | No | 2.355375 |
| Obstacle circular edge expansion grid map | Average value | 1.822855 | 1.259968 | 0.955464 | 0.258427 | 5.052909 | 11.52402 | 20.87364 | 499.4923 |
| | Standard deviation | 0.066401 | 0.193831 | 0.024755 | 0.008742 | 0.361103 | 0.086688 | No | 13.00853 |

paper. Therefore, using the theory and method introduced above, we did the robot motion control experiment, processed the real environment map of the robot motion, planned the safe moving path of the robot, and applied the planning path to the robot motion control.

**3.2.2. Non-standard real environment grid map robot motion control experiment.** In order to carry out ant colony algorithm path planning, we need to set the initial position and target position by Fig 19. Because 4 coding maps have a one-to-one correspondence, if we want to indicate the location in the real environment map, we can indicate it in Fig 19d according to the code number.

According to Fig 19d, specify the initial position and target position as 3229 and 489, then plan the path and pose according to the ant colony algorithm. The planning results are shown in Fig 20a. According to the obstacle recognition results, we get that the size of the reference object in the system is (148 + 149)/2, that is, 148.5, and the actual size of the reference object is 200 mm. Therefore, according to Eq (7), BLC = 148.5/200 = 0.7425.

Combined with the scale, we combine the planning path and pose with the real environment map, and obtain the planning path, planning pose in the real environment. According to the pose planning diagram in Fig 20a, we obtained the motion control coding diagram of the first figure in Fig 22D, and carried out the robot motion control experiment through the motion control coding diagram to obtain the blue trajectory in Fig 22A. From Fig 22A, we can see that the real trajectory of the robot is in contact with the obstacle, because the structure of the robot itself is 180 × 180 mm, and the safety distance of the robot should be greater than 90 mm according to the path passing from the center of the robot. Therefore, the path planning without considering the safety distance of the robot is dangerous for the robot to go from the initial position to the goal position. This is not possible in practice.

**3.2.3. Robot motion control experiment under obstacle circular design grid map.** With the same generation process, we obtained the motion control coding diagram of the second figure in Fig 22D. The experiment result is shown in Fig 22B. The path in Fig 22B is very different from the path in Fig 22A. But such a path is still unsafe, and it can be seen that the path is too close to the obstacle in the center of the map.

**3.2.4. Robot motion control experiment under obstacle circular edge expansion design grid map.** In order to realize the safe planning trajectory of robot motion, and verify the effectiveness of path and pose planning, we have carried out path and pose planning experiments in the original non-standard real environment map and obstacle circular design environment map, and analyzed the planning path and pose. In order to ensure the safety of the planning path and pose, we continue to add the safety distance on the basis of the circular design of the obstacle, and expand the design on the circular edge of the obstacle. Fig 22C shows the motion control test trajectory designed by the circular

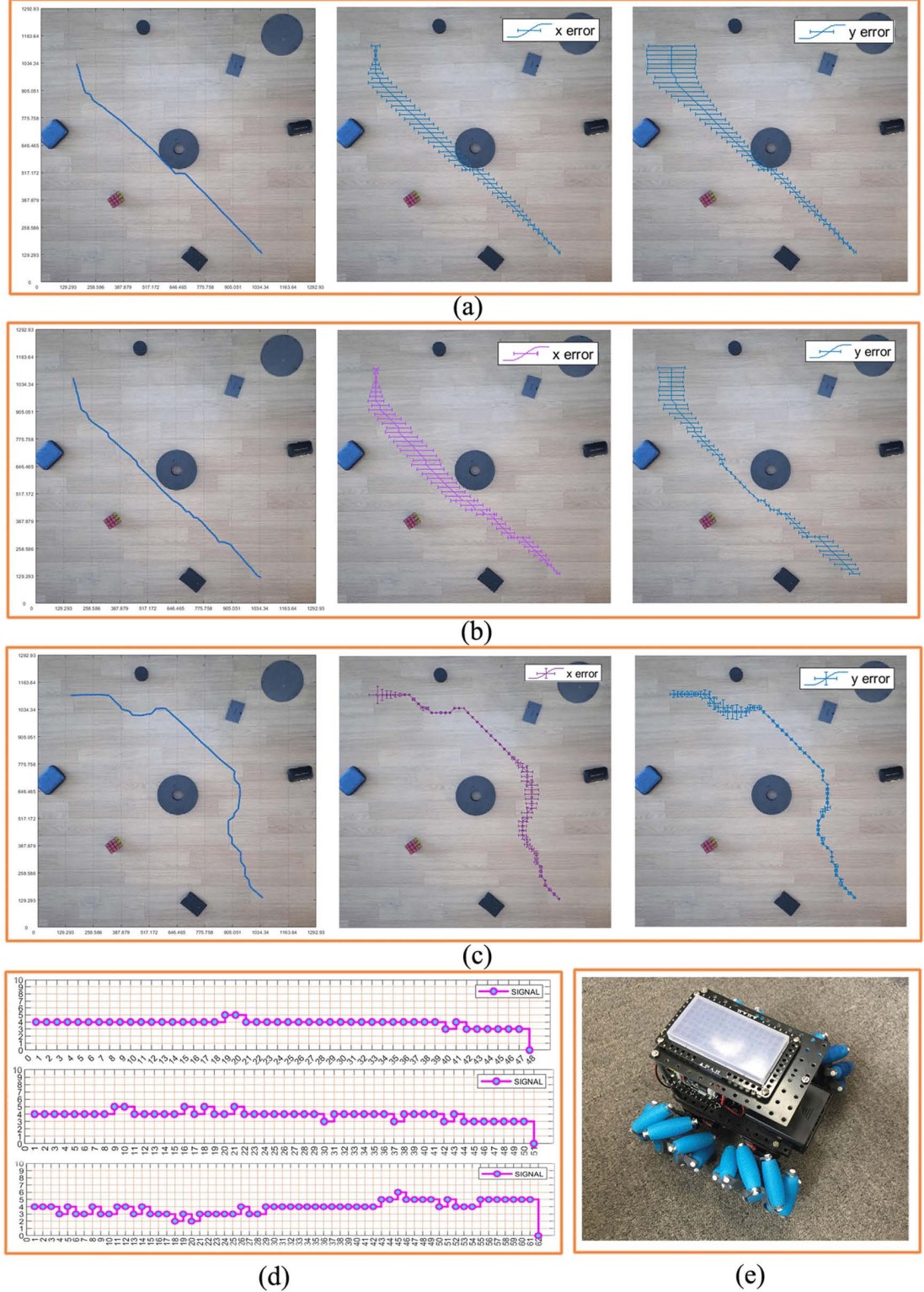

**Fig 21. Real environment planning path security. (a)** Real environment grid map planning path security. **(b)** Circular design of real environment obstacles grid map planning path security. **(c)** Real environment obstacle circular edge expansion design grid map planning path security.

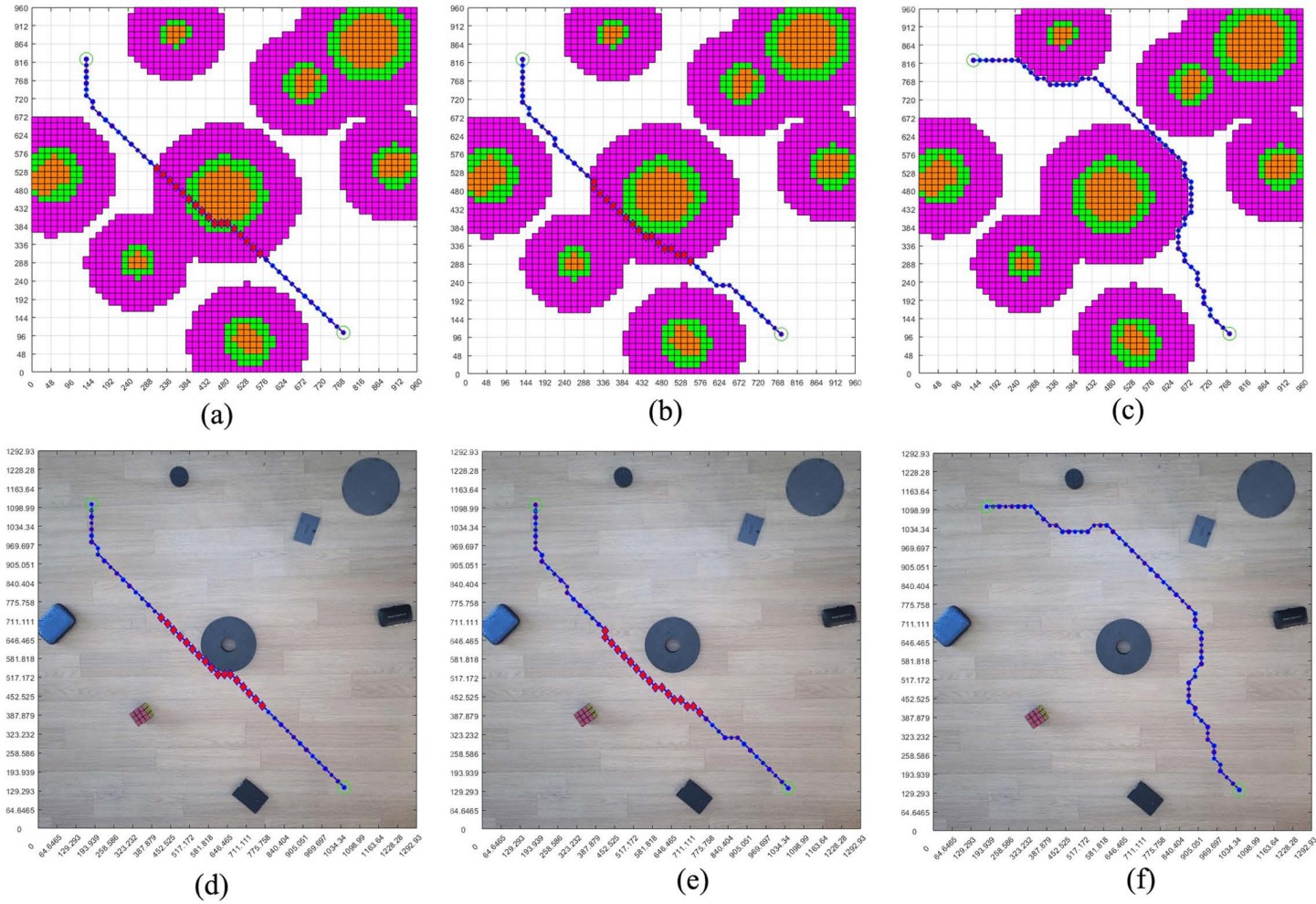

**Fig 22. Real environment robot motion control experiments. (a)** Real environment grid map planning path motion control experiment actual trajectory, x direction error, y direction error. **(b)** Experiment actual trajectory, x direction error, y direction error of the real environment obstacle circular design grid map planning path motion control experiment. **(c)** Experiment actual trajectory, x direction error, y direction error of the real environment obstacle circular edge expansion design grid map planning path motion control experiment. **(d)** Coding graphs under three types of maps. **(e)** Experiment robot.

edge expansion. The third figure of Fig 22D shows the coding graph of obstacle circular edge expansion design grid map path planning. The blue trajectory in Fig 22C is the actual trajectory. It can be seen from the figure that by expanding the circular area of the obstacle, the distance between the planning path and the obstacle is effectively increased, which theoretically ensures that the robot will not contact with the obstacle during the moving process. The circular edge expansion design method of non-standard real environment obstacles can theoretically obtain the planning path and pose of the robot. From the actual motion trajectory, it can be seen that the robot can safely reach the target position from the initial position according to the planning path and pose. During the experiment, the robot does not contact or collide with obstacles and completes the whole process safely.

### 3.2.5. Analyzing the results of the experiment.
As we can see, Fig 22 makes a comparative analysis of the experiment as a whole, and comprehensively investigates the rationality, effectiveness, and safety of the path planning in the real environment, focusing on the error between the planning path and the actual motion trajectory, and the possibility of realizing motion control.

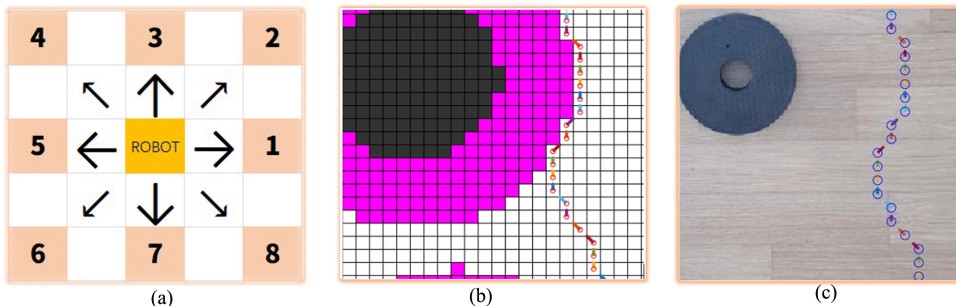

**Fig 23. Robot motion pose encoding. (a)**The encoding value corresponding to the pose. **(b)**The pose in the local grid map. **(c)**The pose in the local real environment map.

Fig 22A shows the motion errors in x and y directions of the actual motion path planned without circularity design, Fig 22B shows the motion errors in x and y directions of the actual motion path planned with circularity design but without expanding the edge of circularity design. Fig 22C shows the motion error in the x and y direction of the actual motion path of the path planning by circular design and circular edge expansion design.

Fig 24 makes a comparative analysis of the experiment as a whole, and comprehensively investigates the rationality, effectiveness and safety of the path planning in the real environment. In Fig 24, the path planning iteration graphs of the three methods are shown in Fig 24a, and the path lengths under the three methods are obtained. As can be seen from the path comparison diagram in Fig 24b, after the obstacle circular design and circular edge expansion design are carried out on the original non-standard real environment map, the length of the planning path of the ant colony algorithm becomes longer, and the obstacle circular edge expansion design has the longest planning path in the grid map.

This is mainly because after the circular design and circular edge expansion design of the obstacles, the obstacles become larger, and the obstacles need to be bypassed when the planning path meets them. However, it is these increases that improve the safety of the planning path on the other hand.

It can be seen from Fig 24c that the planning path in the grid map of the circular edge expansion design of the obstacles, the total number of waypoints is 62. The number of secure waypoints is 62. The number of dangerous waypoints is 0. So the security level isreaching 100%. This is very important for the subsequent motion control of the robot, because safety comes first. After comparison and analysis, it can be concluded that our proposed method is reasonable. Theoretically, through the circular edge expansion design of obstacles, and then the path planning method of ant colony algorithm can not only obtain a safe path, but also use the algorithm itself to optimize the path, and obtain the optimal planning path and pose.

After analysis, in the actual movement of the robot, the actual distance of the robot operation changes due to the circular friction, which also points out a new direction for our future research: how to reduce the change of the error of the operation position caused by the abnormal operation process of the robot and the different friction coefficients everywhere on the circle. However, it can be seen from the Figs 22, 24 and the additional experiments (S1–S3 Figs) that the path planning method proposed in this paper for the circular edge expansion design of obstacles in the real environment is reasonable, effective, safe, and able to realize the motion control of the robot.

### 3.3. Other experiments

**3.3.1. Other motion control experiments.** Fig 25 shows the motion control experiments under different environments. The figure displays the planning path and the actual motion trajectory. Table 7 presents the motion control

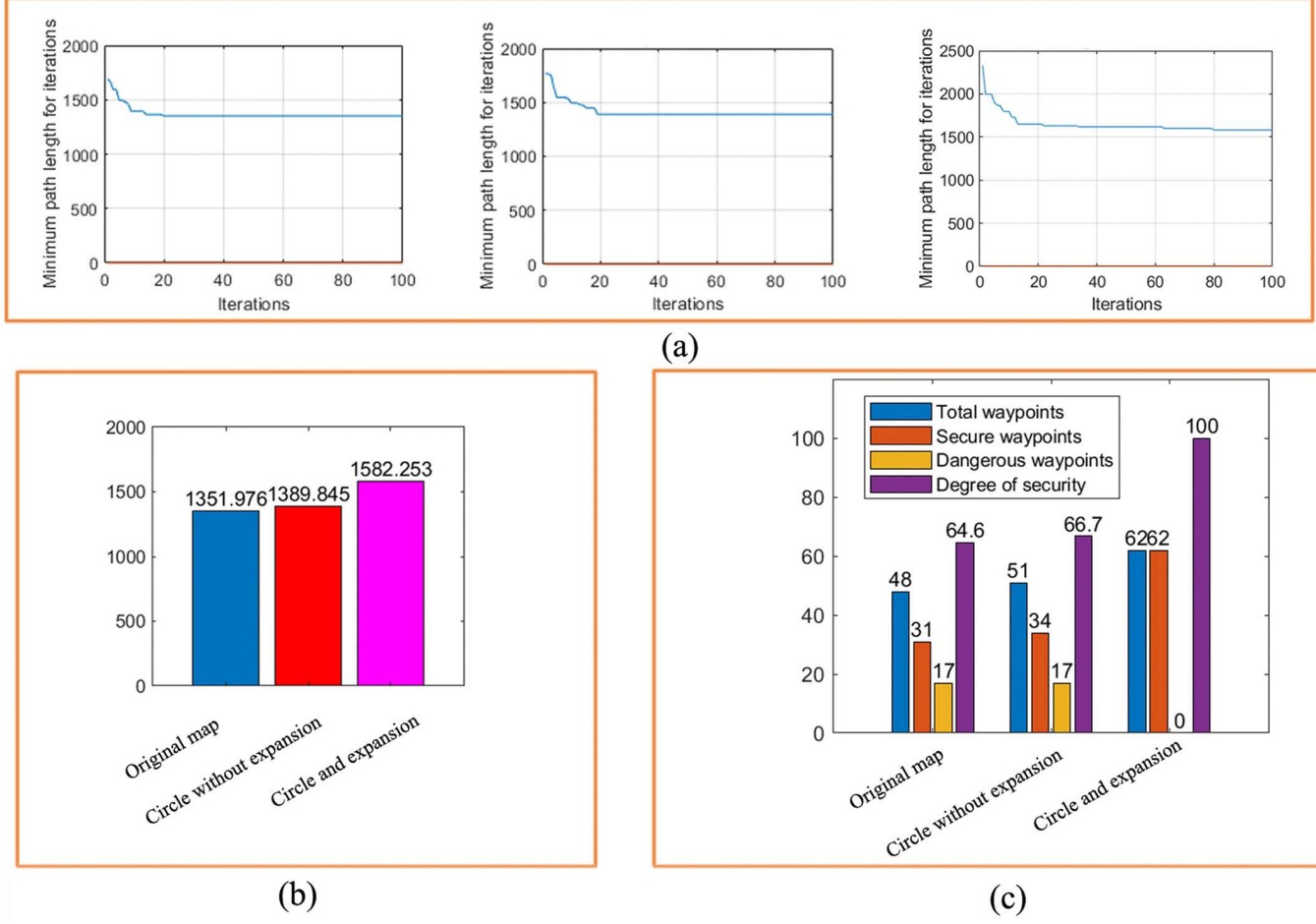

**Fig 24. Real environment three maps planning path security comparison. (a)** Three kinds of map path planning iteration diagrams. **(b)**Changes of planning path length under the three maps. **(c)**Safety degree change graph.

result parameters under several environments. From Fig 24 and Table 7, it can be seen that the motion control of the robot can be achieved by using the method proposed in this paper.

**3.3.2. Other algorithm path planning.** We employed the method of circular edge expansion design of the obstacles and conducted path planning experiments in real environments using other algorithms. The results are shown in Fig 26 and Table 8. From Fig 26 and Table 8, it can also be seen that the application prospects of the method proposed in this paper for path planning by circularly expanding obstacles are extremely broad.

**3.3.3. Other experiments path planning.** In order to verify the effectiveness of the proposed path planning method in realenvironments, we conducted a large number of path planning experiments. As shown in Fig 27, the planning path successfully bypasses the obstacles, reaches the target position from the initial position, and maintains an appropriate distance from the obstacles.

## 4. Discussion

A method of robot path planning based on circular expansion design of obstacles in the real environment map is proposed, which can greatly improve the intelligence of robot path planning. This method is shown in the virtual environment

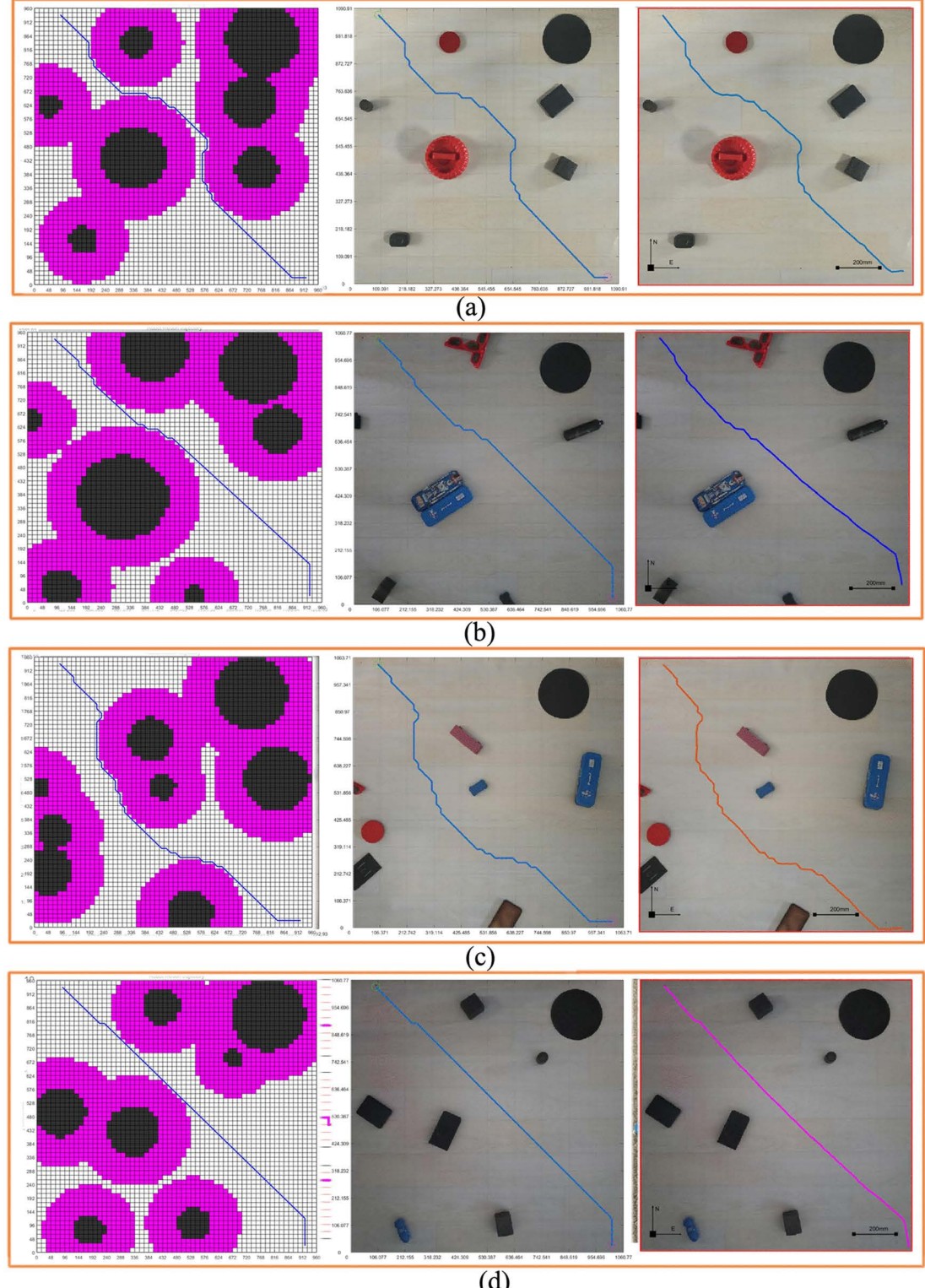

**Fig 25. Different environments grid map planning path, real environment path, and real experiment trajectory. (a)** Experiment 1. **(b)**Experiment 2. **(c)**Experiment 3. **(d)**Experiment 4.

**Table 7. Experimental results of motion control in different environments.**

| Environment number | Initial position encoding | Target position encoding | Minimum gap(mm) | Planning path length(mm) | Actual running path length(mm) | Collision situation |
|---|---|---|---|---|---|---|
| 1 | 66 | 3538 | 185 | 1.560203e + 03 | 1521.294 | No collision |
| 2 | 66 | 3538 | 172 | 1.429962e + 03 | 1390.984 | No collision |
| 3 | 66 | 3538 | 132 | 1.552462e + 03 | 1576.262 | No collision |
| 4 | 66 | 3538 | 139 | 1.398893e + 03 | 1425.224 | No collision |

**Table 8. Experimental results of motion control under different algorithms in real environments.**

| Dijkstra algorithm | PRM algorithm | RRT algorithm | A* algorithm | RPF algorithm | GA algorithm |
|---|---|---|---|---|---|
| 1.1538e + 03 mm | 1.0832e + 03 mm | 1.2575e + 03 mm | 1.1538e + 03 mm | 1.4013e + 03 mm | 1.0730e + 03 mm |

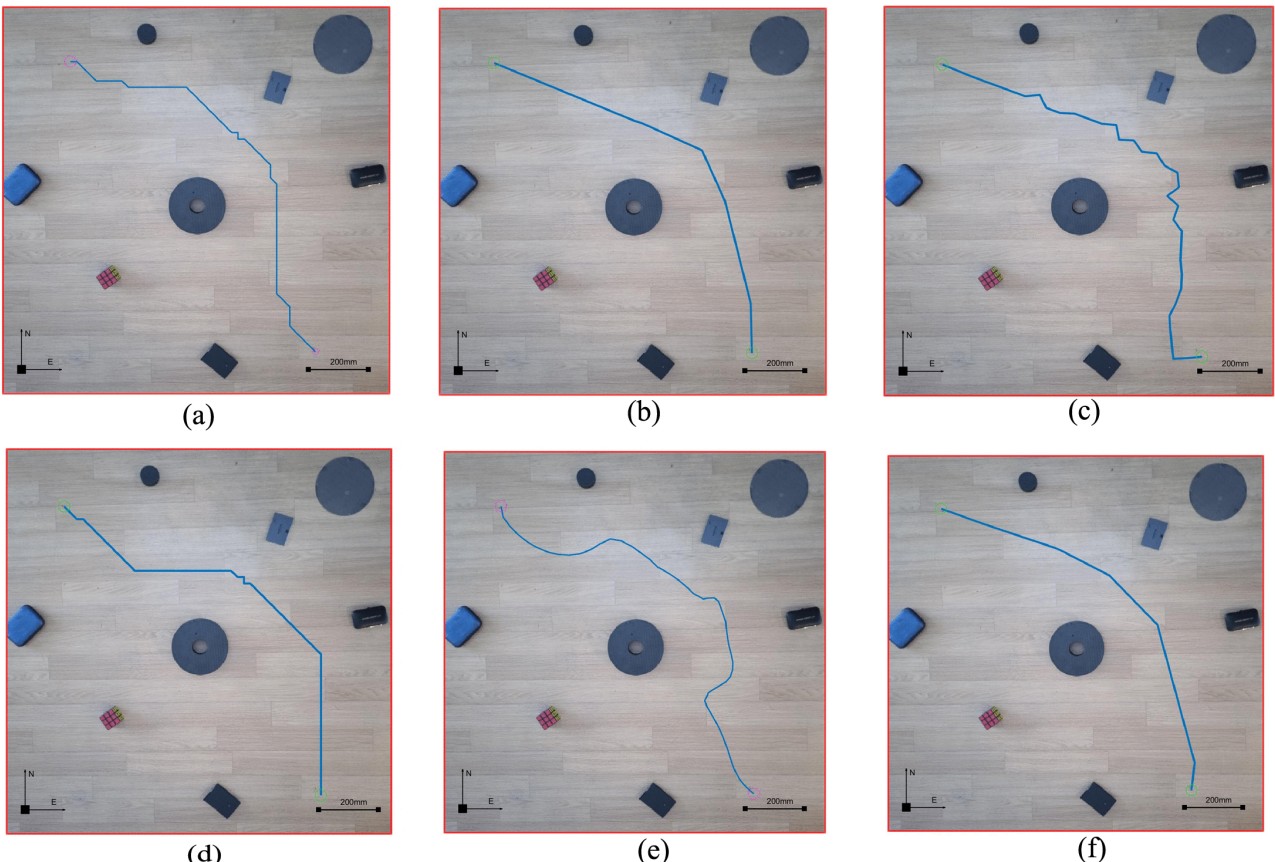

**Fig 26. Display of planning paths under different algorithms in real environments. (a)** Dijkstra algorithm. **(b)** PRM algorithm. **(c)** RRTalgorithm. **(d)** A* algorithm. **(e)** APF algorithm. **(f)** GA algorithm.

  

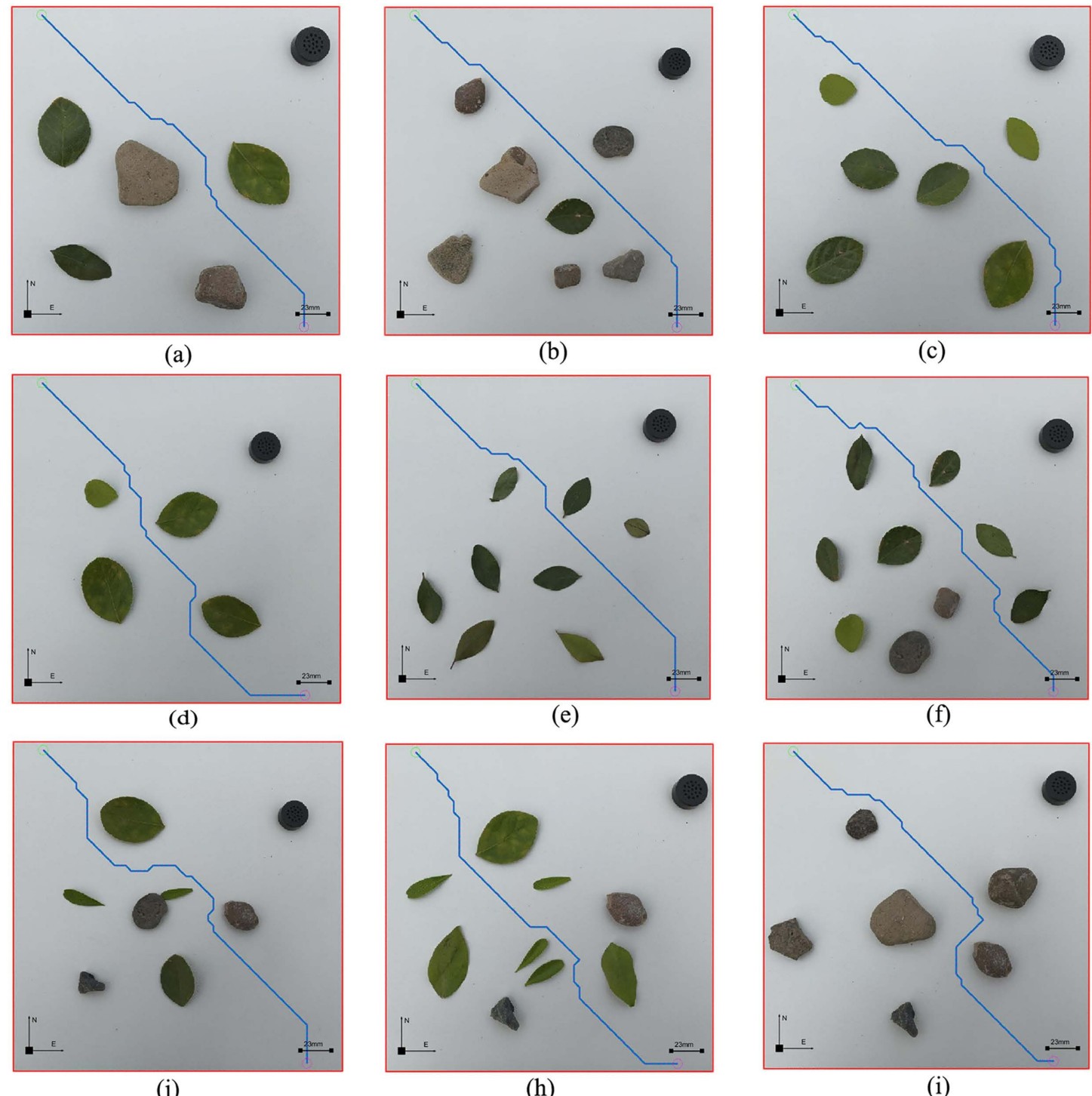

**Fig 27. Display of planning paths in different environments. (a)** Environment 1. **(b)** Environment 2. **(c)** Environment 3. **(d)** Environment 4. **(e)** Environment 5. **(f)** Environment 6. **(j)** Environment 7. **(h)** Environment 8. **(i)** Environment 9.

map and verified in the real environment map. Through the standardization process, the grid method is used to divide the map, and the binarization process is carried out to obtain the standard grid map. The obstacle circular edge expansion design algorithm is used to design the obstacle in the map, and the edge of the obstacle is extended, and the safety distance is added, so that the safety of the generated planning path is improved to 100%. Ant colony algorithm is used to carry out path planning in the generated grid map, and the planning path and planning pose of the robot in the real environment map are obtained, which lays a foundation for the convenience of robot motion control and the improvement of robot intelligence.

This approach can be applied to path planning in different environments. In the real environment, the shape and size of obstacles are different, and the environment of obstacles is also different. As a method of studying path planning in real environment, this method provides technical support for robot motion control to enable robots to deal with various complex scenes. The adaptability of robots to different environments is very important to improve the intelligence of robots. Robots with this real environment path planning ability can quickly deal with logistics and handling work in different environments in industrial production, greatly reducing the labor intensity of programmers and improving the production cycle of factories.

In this method, the safety distance is added to the obstacle edge by using the obstacle circularization method, which ensures the safety of the planning path. The method is combined with the real environment map and displayed in the real environment map. Compared with the traditional expression of planning path and planning pose in the standard grid map, it is more intuitive and promotes the development of robot motion control technology. Because the path planning is carried out in the real environment map with the real object as the reference, the planning pose and path can be directly applied to the robot movement, which greatly facilitates the motion control of the robot.

Finally, this paper combined the virtual environment and real environment to do experimental analysis and research, the path and pose of the robot are planned and generated in the virtual environment, and the real environment, and the robot motion control coding graph is given. The robot motion control experiment is verified in the real environment. Although the path and pose planning is carried out in the virtual environment and the real environment, the planning path and pose are reasonable and safe, and the physical motion control is also carried out through the robot, which achieves the purpose of the research. Because of the practicability and intelligence of the method, its application prospects and application scope will be greatly expanded and continue to improve.

## Supporting information

**S1 Fig. Additional path planning (489–3229).**
(TIF)

**S2 Fig. Additional path planning (3252−157).**
(TIF)

**S3 Fig. Additional path planning (157–3252).**
(TIF)

## Author contributions

**Conceptualization:** Feng Li.

**Data curation:** Maoya Yang.

**Formal analysis:** Feng Li, Seong-Nam Jo.

**Funding acquisition:** Feng Li.

**Investigation:** Young-Chul Kim, Ziang Lyu.

                                    

**Project administration:** Feng Li.

**Software:** Feng Li.

**Supervision:** Seong-Nam Jo, Young-Chul Kim.

**Validation:** Ziang Lyu.

**Visualization:** Feng Li, Maoya Yang.

**Writing – original draft:** Feng Li.

**Writing – review & editing:** Young-Chul Kim.

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
