## [Decision Letter · Decision Letter 0]

14 Sep 2025

PONE-D-25-39430Real environment obstacle circular edge expansion desion robot path planning based on ant colony algorithmPLOS ONE

Dear Dr. Li,

Thank you for submitting your manuscript to PLOS ONE. After careful consideration, we feel that it has merit but does not fully meet PLOS ONE’s publication criteria as it currently stands. Therefore, we invite you to submit a revised version of the manuscript that addresses the points raised during the review process.

The main concerns relate to the **limited scope of the experimental validation, insufficient quantitative analysis, and lack of methodological transparency**. At present, the real-world experiments rely on a single tabletop scenario with one start–goal pair, which is not adequate to demonstrate robustness. The reported results are primarily qualitative, and critical numerical data such as medians, error distributions, runtime decompositions, and sensitivity analyses are missing. Furthermore, the evaluation relies exclusively on ACO as the planning method, without comparison to established baselines such as A*, D*, or RRT*, making it difficult to assess the true merit of the proposed approach.

In addition, several aspects of the methodology are insufficiently described or inconsistently presented, including the image binarization process, coordinate conventions, calibration of clearance thresholds, and the details of the control system used for trajectory execution. These omissions hinder both the interpretability and reproducibility of the study. The manuscript also requires significant improvements in language quality, figure presentation, and data availability, as the current form falls short of journal expectations for clarity and reproducibility.

We look forward to receiving your revised manuscript.

Kind regards,

Francesco Visentin

Academic Editor

PLOS ONE

Journal Requirements:

“This research was funded by the Science and Technology Research Project of Henan Province, China (No. 222102210307).”

Additional Editor Comments :

Reviewer #1:

1- The real-world evaluation appears to rely on a single tabletop layout (7 obstacles + 1 reference) captured by an overhead phone, with one chosen start–goal pair (codes 66 → 3538). This is not sufficient to establish robustness. Please repeat the full pipeline on multiple real environments (vary obstacle densities/shapes, start–goal pairs, and robot sizes) and report aggregate statistics (success rate, min-clearance, path length, tracking error, collisions). The current description shows one map being standardized (Fig. 13–14/16–18) and then used for the three comparative cases only.

2- Fig. 21/22 present x/y error and pose error “box diagrams,” but the manuscript reports only qualitative impressions (e.g., “median generally lower than 5°,” “y-error about 8”). Please tabulate the exact medians, IQRs, and RMS/MAE for x, y, and pose, and clarify how “abnormal points” (outliers) were identified/removed. Without explicit numbers and outlier policy, the comparison across the three maps is hard to interpret.

3- In the virtual-environment test, the authors note “17 dangerous points” along the planned pose before expansion; that’s useful, but please (i) state the exact δ used to classify danger per Eq. (19), (ii) report the count/min-clearance for all three maps in both virtual and real experiments, and (iii) provide distributions of point-to-obstacle distances along the final paths. A small summary table would make the improvement unambiguous.

4- You report “from original map to Fig. 18 used 23.075115 s.” Please decompose this into per-stage times (binarization, obstacle detection, circularization, edge expansion, gridding, ACO planning), list hardware/OS and software versions, and provide averages ± SD over ≥10 runs. Also report ACO convergence iterations/time for each of the three maps.

5- All comparisons hold the planner fixed (ACO) and vary map preprocessing (raw vs circular vs circular+expansion). For completeness, please add at least one classical baseline (e.g., A*, D*, RRT/RRT*) on the same three maps and report the same metrics (path length, min-clearance, collisions, planning time). This will show whether the claimed safety/robustness benefits persist beyond a single metaheuristic. (This request is motivated by the paper’s exclusive reliance on ACO in the current Results.

6- The “time coding graph” approach implies mostly open-loop pose execution; please detail the controller (kinematics of the four omni wheels, sampling period, speed limits, any odometry/IMU/vision feedback), the error computation method (in what frame? units on axes?), and how the time codes were generated from poses. These details are essential to interpret the tracking error plots in Fig. 21/22 and to reproduce the experiments. Also state whether wheel slip/friction was characterized.

7- BLC is computed from a 200 mm reference (REDMI 9A overhead capture), yielding BLC = 0.7425. Please report calibration uncertainty (lens distortion, perspective, height variation), how BLC propagates to path-length and clearance errors, and whether the min-clearance threshold δ includes a calibration safety margin. Without this, the mapping from pixels to millimeters in the Results remains uncertain.

Reviewer #2:

Thanks for sharing—there’s a very useful idea here. That said, in its current form it needs a major revision.

1.The author define J = floor(B/255) and then explain “<255→0, =255→1” (L158–L162). That basically treats only pure white as free space—too brittle. Use a real threshold (Otsu or EDT→threshold), report T and robustness.

2.The author first set origin at top-left, +x right, +y down (L147–L149, L172–L174), but later you switch to left→right & bottom→top (L191). Pick one convention and rewrite the encode/decode formulas accordingly.

3.The author state the rule Lij ≤ δ (unsafe) and Lij > δ (safe) (L269 & L273), and you expand obstacles with Rsi = (LCol + LCow)/2 + Ls (L305–L307), but there’s no calibration of δ / Ls (units, how chosen, sensitivity). Define δ = robot radius + sensing/pose error margin, implement via configuration-space dilation / EDT, and show a quick sensitivity check.

4.The author list ACO params only (K=100, M=100, α=1, β=30, ρ=0.2, Q=100) (≈L343–L351), claim 23.075115 s end-to-end (L427), but give no HW/SW context or manual/auto breakdown, no baselines (A*/D* Lite/RRT*), and later show a “Safety degree change graph” (L533) without a mathematical definition. Report path length, minimum clearance, runtime (mean±SD/CI), add baselines, define “safety” formally, and back the claim with stats.

Reviewer #3:

The contribution here is very incremental, obstacle expansion and ant colony optimization are not really new ideas. The experiments are small scale and very controlled, no replication or solid statistical robustness is shown. Also no comparisons with standard baselines (A*, D*, RRT etc) so it’s hard to judge if this method is actually effective. The claim of “100% safety” is overstated, not supported with strong evidence. The language quality is weak, lot of grammatical errors, awkward phrasing, even typos. Figures feel repetitive and captions don’t explain clearly. Data availability is also not enough for reproducibility, no raw trajectories, code or scripts provided. Overall, the presentation falls short of journal standards.

Reviewers' comments:

Reviewer's Responses to Questions

**Comments to the Author**

1. Is the manuscript technically sound, and do the data support the conclusions?

Reviewer #1: Yes

Reviewer #2: Yes

Reviewer #3: Partly

2. Has the statistical analysis been performed appropriately and rigorously? 

Reviewer #1: Yes

Reviewer #2: Yes

Reviewer #3: No

3. Have the authors made all data underlying the findings in their manuscript fully available?

Reviewer #1: Yes

Reviewer #2: Yes

Reviewer #3: Yes

4. Is the manuscript presented in an intelligible fashion and written in standard English?

Reviewer #1: Yes

Reviewer #2: Yes

Reviewer #3: No

5. Review Comments to the Author

Reviewer #1: 1- The real-world evaluation appears to rely on a single tabletop layout (7 obstacles + 1 reference) captured by an overhead phone, with one chosen start–goal pair (codes 66 → 3538). This is not sufficient to establish robustness. Please repeat the full pipeline on multiple real environments (vary obstacle densities/shapes, start–goal pairs, and robot sizes) and report aggregate statistics (success rate, min-clearance, path length, tracking error, collisions). The current description shows one map being standardized (Fig. 13–14/16–18) and then used for the three comparative cases only.

2- Fig. 21/22 present x/y error and pose error “box diagrams,” but the manuscript reports only qualitative impressions (e.g., “median generally lower than 5°,” “y-error about 8”). Please tabulate the exact medians, IQRs, and RMS/MAE for x, y, and pose, and clarify how “abnormal points” (outliers) were identified/removed. Without explicit numbers and outlier policy, the comparison across the three maps is hard to interpret.

3- In the virtual-environment test, the authors note “17 dangerous points” along the planned pose before expansion; that’s useful, but please (i) state the exact δ used to classify danger per Eq. (19), (ii) report the count/min-clearance for all three maps in both virtual and real experiments, and (iii) provide distributions of point-to-obstacle distances along the final paths. A small summary table would make the improvement unambiguous.

4- You report “from original map to Fig. 18 used 23.075115 s.” Please decompose this into per-stage times (binarization, obstacle detection, circularization, edge expansion, gridding, ACO planning), list hardware/OS and software versions, and provide averages ± SD over ≥10 runs. Also report ACO convergence iterations/time for each of the three maps.

5- All comparisons hold the planner fixed (ACO) and vary map preprocessing (raw vs circular vs circular+expansion). For completeness, please add at least one classical baseline (e.g., A*, D*, RRT/RRT*) on the same three maps and report the same metrics (path length, min-clearance, collisions, planning time). This will show whether the claimed safety/robustness benefits persist beyond a single metaheuristic. (This request is motivated by the paper’s exclusive reliance on ACO in the current Results.

6- The “time coding graph” approach implies mostly open-loop pose execution; please detail the controller (kinematics of the four omni wheels, sampling period, speed limits, any odometry/IMU/vision feedback), the error computation method (in what frame? units on axes?), and how the time codes were generated from poses. These details are essential to interpret the tracking error plots in Fig. 21/22 and to reproduce the experiments. Also state whether wheel slip/friction was characterized.

7- BLC is computed from a 200 mm reference (REDMI 9A overhead capture), yielding BLC = 0.7425. Please report calibration uncertainty (lens distortion, perspective, height variation), how BLC propagates to path-length and clearance errors, and whether the min-clearance threshold δ includes a calibration safety margin. Without this, the mapping from pixels to millimeters in the Results remains uncertain.

Reviewer #2: Thanks for sharing—there’s a very useful idea here. That said, in its current form it needs a major revision.

1.The author define J = floor(B/255) and then explain “<255→0, =255→1” (L158–L162). That basically treats only pure white as free space—too brittle. Use a real threshold (Otsu or EDT→threshold), report T and robustness.

2.The author first set origin at top-left, +x right, +y down (L147–L149, L172–L174), but later you switch to left→right & bottom→top (L191). Pick one convention and rewrite the encode/decode formulas accordingly.

3.The author state the rule Lij ≤ δ (unsafe) and Lij > δ (safe) (L269 & L273), and you expand obstacles with Rsi = (LCol + LCow)/2 + Ls (L305–L307), but there’s no calibration of δ / Ls (units, how chosen, sensitivity). Define δ = robot radius + sensing/pose error margin, implement via configuration-space dilation / EDT, and show a quick sensitivity check.

4.The author list ACO params only (K=100, M=100, α=1, β=30, ρ=0.2, Q=100) (≈L343–L351), claim 23.075115 s end-to-end (L427), but give no HW/SW context or manual/auto breakdown, no baselines (A*/D* Lite/RRT*), and later show a “Safety degree change graph” (L533) without a mathematical definition. Report path length, minimum clearance, runtime (mean±SD/CI), add baselines, define “safety” formally, and back the claim with stats.

Reviewer #3: The contribution here is very incremental, obstacle expansion and ant colony optimization are not really new ideas. The experiments are small scale and very controlled, no replication or solid statistical robustness is shown. Also no comparisons with standard baselines (A*, D*, RRT etc) so it’s hard to judge if this method is actually effective. The claim of “100% safety” is overstated, not supported with strong evidence. The language quality is weak, lot of grammatical errors, awkward phrasing, even typos. Figures feel repetitive and captions don’t explain clearly. Data availability is also not enough for reproducibility, no raw trajectories, code or scripts provided. Overall, the presentation falls short of journal standards.

6. PLOS authors have the option to publish the peer review history of their article (what does this mean?). If published, this will include your full peer review and any attached files.

Reviewer #1: No

Reviewer #2: No

Reviewer #3: No

---

## [Author Response · Author response to Decision Letter 1]

28 Feb 2026

Reviewer #1:

1- The real-world evaluation appears to rely on a single tabletop layout (7 obstacles + 1 reference) captured by an overhead phone, with one chosen start–goal pair (codes 66 → 3538). This is not sufficient to establish robustness. Please repeat the full pipeline on multiple real environments (vary obstacle densities/shapes, start–goal pairs, and robot sizes) and report aggregate statistics (success rate, min-clearance, path length, tracking error, collisions). The current description shows one map being standardized (Fig. 13–14/16–18) and then used for the three comparative cases only.

Dear reviewer, thank you for your review. To address your comments, we have added the path planning and motion control experiments (as shown in Fig 25 and Table 7). Additionally, we have also added the path planning in different non-standard environments and demonstrated the path planning results (as shown in Fig 27). Since this paper focuses on the path planning in non-standard real environments, the robot's path tracking mainly studies the control accuracy of the robot system, and is not the end point of our research. The path planning in different environments can precisely illustrate the effectiveness of the methods proposed in this paper. Thank you for your guidance. In our subsequent research, we will conduct more in-depth studies on the robot itself.

Please see: (Response to reviewers).

2- Fig. 21/22 present x/y error and pose error “box diagrams,” but the manuscript reports only qualitative impressions (e.g., “median generally lower than 5°,” “y-error about 8”). Please tabulate the exact medians, IQRs, and RMS/MAE for x, y, and pose, and clarify how “abnormal points” (outliers) were identified/removed. Without explicit numbers and outlier policy, the comparison across the three maps is hard to interpret.

Thank you for your suggestion. This paper mainly focuses on path planning using non-standard real-world environment maps. The table is provided merely to demonstrate that the path planning conducted using the methods presented in the paper can be applied to the motion control of robots. We hope to prove the feasibility and effectiveness of the proposed methods through experiments. The research on the accuracy of robot motion control is mainly about the study of robots, rather than the main focus of this paper. Therefore, we did not elaborate much. Please understand. Also, thank you for your reminder. We will conduct further research on the robot system in the future to effectively improve the motion control accuracy of the robots.

3- In the virtual-environment test, the authors note “17 dangerous points” along the planned pose before expansion; that’s useful, but please (i) state the exact δ used to classify danger per Eq. (19), (ii) report the count/min-clearance for all three maps in both virtual and real experiments, and (iii) provide distributions of point-to-obstacle distances along the final paths. A small summary table would make the improvement unambiguous.

Dear reviewer, thank you for your review. To further improve the article, we have added relevant descriptions and provided Figure 21. Please review them. Through these improvements, the research significance of this article has been effectively enhanced. Thank you very much for your guidance.

Please see: (Response to reviewers).

4- You report “from original map to Fig. 18 used 23.075115 s.” Please decompose this into per-stage times (binarization, obstacle detection, circularization, edge expansion, gridding, ACO planning), list hardware/OS and software versions, and provide averages ± SD over ≥10 runs. Also report ACO convergence iterations/time for each of the three maps.

Thank you for your review. Based on your suggestions, we have added Tables 5 and 6. Please review them. Through these improvements, the research significance of this article has been effectively enhanced. Thank you very much for your guidance.

Please see: (Response to reviewers).

5- All comparisons hold the planner fixed (ACO) and vary map preprocessing (raw vs circular vs circular+expansion). For completeness, please add at least one classical baseline (e.g., A*, D*, RRT/RRT*) on the same three maps and report the same metrics (path length, min-clearance, collisions, planning time). This will show whether the claimed safety/robustness benefits persist beyond a single metaheuristic. (This request is motivated by the paper’s exclusive reliance on ACO in the current Results.

Dear reviewer, thank you very much for your guidance. Since path planning using non-standard real-world environment maps is a very large topic, in this paper, only ACO was used for path planning, which demonstrates the feasibility of the proposed method. To further improve the content of this paper, the research team also used maps to conduct path planning with methods such as A*, RRT, and PRM. Since the D* algorithm was not used by the research team, it was not presented. Please refer to Figure 26 and Table 8 for details. These figures and tables illustrate the feasibility and effectiveness of the methods. Due to the different operation methods of various algorithms and the related parameters of other algorithms involving the publication and contribution of other members, it is not convenient to provide them. Please understand.

Please see: (Response to reviewers).

6- The “time coding graph” approach implies mostly open-loop pose execution; please detail the controller (kinematics of the four omni wheels, sampling period, speed limits, any odometry/IMU/vision feedback), the error computation method (in what frame? units on axes?), and how the time codes were generated from poses. These details are essential to interpret the tracking error plots in Fig. 21/22 and to reproduce the experiments. Also state whether wheel slip/friction was characterized.

Dear reviewer, the "time coding graph" is an encoded graph generated from the pose graph to control the movement of the robot. The robot has 8 movement directions in each grid, and each direction is assigned a code, as shown in Figure 22. The time coding graph enables the robot to maintain a signal for a certain duration for each encoded signal. This duration is equal to the length of the grid divided by the robot's movement speed. To better illustrate the principle of the encoding, we have provided a pose encoding graph, as shown in Figure 23d.

The position deviation of the robot's movement is, on the one hand, caused by the accuracy of the robot's movement itself, and on the other hand, caused by the friction and slipping of the robot on the ground when it moves. The robot motion control experiments in the text are designed to verify the feasibility and effectiveness of the proposed planning path using the non-standard environmental map in the real environment. In fact, the figures 21-27 presented in the text have already demonstrated the feasibility and effectiveness of the planning path. We have verified the planned path and pose using the robot for experiments, which not only enriches the research content of this paper but also re-verifies the feasibility and effectiveness of the proposed path and pose planning in the real environment using non-standard environmental maps.

The items you mentioned (kinematics of the four omni wheels, sampling period, speed limits, any odometry/IMU/vision feedback) are mainly about the research of the robot. We have conducted relatively few studies on the robot itself, and our research mainly focuses on its applications. Thank you for your reminder. We will also conduct an in-depth study on the robot itself in the future.

Please see: (Response to reviewers).

7- BLC is computed from a 200 mm reference (REDMI 9A overhead capture), yielding BLC = 0.7425. Please report calibration uncertainty (lens distortion, perspective, height variation), how BLC propagates to path-length and clearance errors, and whether the min-clearance threshold δ includes a calibration safety margin. Without this, the mapping from pixels to millimeters in the Results remains uncertain.

Dear reviewer, thank you for your review. The issues you raised have been addressed and explained in the text as follows. It is precisely because of the calibration uncertainty that we included the safety distance when calculating the threshold δ. In fact, we can directly observe the rationality of the planned path and its effectiveness from Figures 20, 21, 25, and 27. The path planning effect can be clearly seen from the figures, and it is very good. Thank you for your reminder. We will also conduct relevant research on this issue in the future to improve the path planning effect. Since path planning in real environments is a very large topic involving many research directions, thank you for your guidance. We will continue to improve and optimize in the future.

Please see: (Response to reviewers).

Reviewer #2:

Thanks for sharing—there’s a very useful idea here. That said, in its current form it needs a major revision.

1.The author define J = floor(B/255) and then explain “<255→0, =255→1” (L158–L162). That basically treats only pure white as free space—too brittle. Use a real threshold (Otsu or EDT→threshold), report T and robustness.

Dear reviewer, thank you very much for your valuable comments. Utilizing J = floor(B/255) to binarize the matrix is a common method for converting a map into binary form. Each researcher can set it according to their own requirements. You can also use the ROUND function, but usually this method is employed. We can further expand the boundary of the obstacles by adding a safety distance and other methods to ensure the safety of the planned path.

Dividing the obstacle boundaries using a computer is relatively accurate. In fact, if the computer cannot distinguish the boundaries, in real scenarios, it is even more difficult for the human eye to distinguish the boundaries of the obstacles. Since the method proposed in this paper is for path planning in non-standard real environments, and the topic of path planning in non-standard real environments is very large, there are many subsequent studies that need to be conducted, including the method you proposed for distinguishing the boundaries of the obstacles. Thank you very much for your guidance. Your guidance has also provided us with a new research direction, and our research group will continue to explore in this direction in the future.

2.The author first set origin at top-left, +x right, +y down (L147–L149, L172–L174), but later you switch to left→right & bottom→top (L191). Pick one convention and rewrite the encode/decode formulas accordingly.

Dear reviewer, thank you for your review. We didn't notice this issue in the early stage. Therefore, we have improved all the figures in the article, agreed on the setting principles, and the improvements can be seen in the figures within the article.

Please see: (Response to reviewers).

3.The author state the rule Lij ≤ δ (unsafe) and Lij > δ (safe) (L269 & L273), and you expand obstacles with Rsi = (LCol + LCow)/2 + Ls (L305–L307), but there’s no calibration of δ / Ls (units, how chosen, sensitivity). Define δ = robot radius + sensing/pose error margin, implement via configuration-space dilation / EDT, and show a quick sensitivity check.

Dear reviewer, thank you for your review. The setting of δ has been described in the text. Because δ has already been incorporated into the map during its generation, that is, it has expanded the circle. We deformed the obstacles in this way to ensure that the safety distance was included during the design process. Therefore, this δ is not for checking purposes, but is a requirement that needs to be met when designing the obstacles. We demonstrated the planned path in the real environment and verified the effectiveness of the planned path again.

Please see: (Response to reviewers).

4.The author list ACO params only (K=100, M=100, α=1, β=30, ρ=0.2, Q=100) (≈L343–L351), claim 23.075115 s end-to-end (L427), but give no HW/SW context or manual/auto breakdown, no baselines (A*/D* Lite/RRT*), and later show a “Safety degree change graph” (L533) without a mathematical definition. Report path length, minimum clearance, runtime (mean±SD/CI), add baselines, define “safety” formally, and back the claim with stats.

Dear reviewer, thank you for your review. In the article, it has been pointed out in the forefront that currently, the main methods are to obtain real environmental data through self-modeling or through LiDAR, and then model based on the real environmental data. Therefore, there are difficulties in path planning. If only using a non-standard real environmental map for path planning, the difficulty will be greatly reduced and the intelligence level will be greatly improved. This is the significance of this article, and the safety change graph of this article is also one of the research methods explained in this article. Since there is no reference for comparison, there are no baselines as you mentioned. To address your question, the research group has made the following supplements in the article:

(1) By conducting path planning in other algorithms (A*/PRM/RRT/GA/Dijkstra/APF), the feasibility and effectiveness of the path planning using non-standard real environmental maps proposed in this article are illustrated.

(2) The concept of safety level was explained, and relevant diagrams were also provided.

(3) According to the explanation, we have presented the statistical results in Figure 24 .

Please see: (Response to reviewers).

Thank you very much for your attention. As the research topic is in a relatively new field, there are insufficient relevant references. However, the proposal of the method in this paper has extremely significant meaning and serves as a valuable reference for the research on global path planning methods in real environments.

Reviewer #3:

The contribution here is very incremental, obstacle expansion and ant colony optimization are not really new ideas. The experiments are small scale and very controlled, no replication or solid statistical robustness is shown. Also no comparisons with standard baselines (A*, D*, RRT etc) so it’s hard to judge if this method is actually effective. The claim of “100% safety” is overstated, not supported with strong evidence. The language quality is weak, lot of grammatical errors, awkward phrasing, even typos. Figures feel repetitive and captions don’t explain clearly. Data availability is also not enough for reproducibility, no raw trajectories, code or scripts provided. Overall, the presentation falls short of journal standards.

Dear reviewer, regarding your review comments, I provide a detailed explanation below. Thank you very much.

（1）“The contribution here is very incremental, obstacle expansion and ant colony optimization are not really new ideas. ”

Dear reviewer, current research mainly takes place in virtual environments or obtains real environmental information through lidar and then builds an environmental map. However, there are few studies on conducting global path planning research using non-standard environmental maps of the real environment. This is the core contribution of this paper. And this has a very significant importance for the research on robot path planning, greatly enhancing the intelligence level of robots.

（2）The experiments are small scale and very controlled, no replication or solid statistical robustness is shown.

Dear reviewer, our research team conducted numerous experiments, but only a small portion of them are presented in the paper. Since the topic of real-world environments versus standard environment maps is a very broad one, this paper theoretically explains it through virtual non-standard environment maps and verifies it through real non-standard environment maps. In response to this comment, we have included Figures 25, 26, and 27 in this paper for partial presentation. These can demonstrate the effectiveness of the method and we hope for your approval.

（3）Also no comparisons with standard baselines (A*, D*, RRT etc) so it’s hard to judge if this method is actually effective.

Dear reviewer, as this paper introduces a method for path planning in a real environment using non-standard real environment maps, this me

---

## [Decision Letter · Decision Letter 1]

20 Apr 2026

Real environment obstacle circular edge expansion design robot path planning based on ant colony algorithm

PONE-D-25-39430R1

Dear Dr. Li,

We’re pleased to inform you that your manuscript has been judged scientifically suitable for publication and will be formally accepted for publication once it meets all outstanding technical requirements.

Kind regards,

Francesco Visentin

Academic Editor

PLOS One

Additional Editor Comments (optional):

After consideration of the revised manuscript, the authors’ rebuttal, and the current reviewer reports, I find that the manuscript has improved substantially relative to the previous round. In particular, the revised version includes additional experiments, timing breakdowns, results across multiple environments, and comparisons with alternative planning algorithms, which help clarify the practical scope of the proposed workflow and address several of the earlier concerns.

Reviewers' comments:

Reviewer's Responses to Questions

**Comments to the Author**

1. If the authors have adequately addressed your comments raised in a previous round of review and you feel that this manuscript is now acceptable for publication, you may indicate that here to bypass the “Comments to the Author” section, enter your conflict of interest statement in the “Confidential to Editor” section, and submit your "Accept" recommendation.

Reviewer #1: (No Response)

Reviewer #3: (No Response)

2. Is the manuscript technically sound, and do the data support the conclusions?

Reviewer #1: (No Response)

Reviewer #3: Yes

3. Has the statistical analysis been performed appropriately and rigorously? 

Reviewer #1: (No Response)

Reviewer #3: Yes

4. Have the authors made all data underlying the findings in their manuscript fully available?

Reviewer #1: (No Response)

Reviewer #3: No

5. Is the manuscript presented in an intelligible fashion and written in standard English?

Reviewer #1: (No Response)

Reviewer #3: Yes

6. Review Comments to the Author

Reviewer #1: Upon reviewing the authors' responses and the revised manuscript, the paper has been significantly improved.

Reviewer #3: Most comments have been addressed to some extent, and the added figures and tables help. But in several places, the response is just “this is not the focus,” which is fair, but it leaves some gaps in how to interpret and trust the results.

They do mention some of these points here and there, but it’s scattered. It would help to clearly state these as limitations in one place, so the scope of the work is obvious.

Maybe make minor changes -

Add simple numbers where comparisons are made, not just figures

Clearly define terms like safety, and avoid strong claims like “100% safety” without context

Briefly describe the experimental setup assumptions where needed

Keep baseline comparisons consistent, even if not very detailed

Be upfront about what is not reproducible (code, data) and why

7. PLOS authors have the option to publish the peer review history of their article (what does this mean?). If published, this will include your full peer review and any attached files.

Reviewer #1: No

Reviewer #3: No

---

## [Editor Report · Acceptance letter]

PONE-D-25-39430R1

PLOS One

Dear Dr. Li,

I'm pleased to inform you that your manuscript has been deemed suitable for publication in PLOS One. Congratulations! Your manuscript is now being handed over to our production team.

Kind regards,

on behalf of

Dr. Francesco Visentin

Academic Editor

PLOS One